



# Using soil water isotopes to infer the influence of contrasting urban green space on ecohydrological partitioning.

Lena-Marie Kuhlemann[1,2], Doerthe Tetzlaff[1,2,3], Aaron Smith[1], Birgit Kleinschmit[4], and Chris Soulsby[3,5,1]

[1]Department of Ecohydrology, Leibniz Institute of Freshwater Ecology and Inland Fisheries, Müggelseedamm 310, 12587 Berlin, Germany
[2]Department of Geography, Humboldt University of Berlin, Rudower Chaussee 16, 12489 Berlin, Germany
[3]Northern Rivers Institute, University of Aberdeen, St. Mary's Building, Kings College, Old Aberdeen, AB24 3UE, Scotland
[4]Institute of Landscape Architecture and Environmental Planning, Technical University Berlin, Straße des 17. Juni 145, 10623 Berlin, Germany
[5]Chair of Water Resources Management and Modeling of Hydrosystems, Technical University Berlin, Gustav-Meyer-Allee 25, 13355 Berlin, Germany

**Correspondence:** Lena-Marie Kuhlemann (kuhlemann@igb-berlin.de)

**Abstract.** Many urban areas are facing challenges in balancing domestic and industrial water demands while simultaneously maintaining the water supply for green infrastructure. Consequently, quantitative knowledge about ecohydrological partitioning in different types of urban green space is crucial for balancing sustainable water needs in cities during future challenges of increasing urbanization and climate warming. Using isotopic tracers in precipitation and soil water, along with conventional

hydrometric measurements in a plot-scale study in Berlin, Germany, we investigated water partitioning under different generic types of urban vegetation (grassland, shrub and trees). This allowed assessment of effects on subsequent evapotranspiration, subsurface flow paths and storage during a prolonged drought period with episodic rainfall. Water losses under forest were slightly higher than grassland over the monitoring in the growing season of 2019. Despite higher soil evaporation losses under urban grassland, higher interception and transpiration likely contributed to slower turnover of soil water and older

groundwater recharge under urban trees. Shrub vegetation seemed to be most resilient to prolonged drought periods, with lower evapotranspiration losses. Our results contribute to a better understanding of ecohydrological partitioning under mixed urban vegetation communities and an evidence base for better adaptive management of urban water and irrigation strategies to sustainably meet the water demands of urban green spaces.

## 1   Introduction

In many cities, hydrology and ecosystem services underpinning sustainable living are closely interlinked (Pataki et al. , 2011a). Besides sustaining domestic and industrial water supplies, urban water resources need to meet the demands of vegetation in a diversity of green spaces, which are increasingly recognized as important "infrastructure" in towns and cities (Nouri et al. , 2013, 2019). This "green infrastructure" provides a range of potential benefits, including enhancement of groundwater recharge, infiltration of flood waters, increased biodiversity, habitat connectivity, and numerous socio-economic benefits (Nouri et al. ,





2013; McGrane , 2016). In terms of ecohydrological partitioning, vegetation cover primarily impacts urban evapotranspiration (*ET*; e.g. Fletcher et al. , 2013; Peters et al. , 2011; Pataki et al , 2011b) and can provide a cooling effect, mitigating urban heat islands (*UHI*; e.g. Gunawardena et al. , 2017). This recognition is contributing to a new paradigm in urban hydrology that seeks to balance traditional engineering needs to evacuate runoff from flood-risk areas, whilst maintaining zones for infiltration and water storage to retain water within built-up areas to underpin beneficial ecosystem services (Fletcher et al. , 2013). Water
resources in many cities are already under increased pressure from climate breakdown, extensive abstractions to maintain supplies for their growing population and economy, ageing infrastructure and increased reliance on water imports, often from long distances (Pataki et al. , 2011a). By 2050, almost 70 % of the world's population is projected to be living in urban areas (UN , 2019), with an expected 80 % increase in urban water demand (Flörke et al. , 2018). Therefore, enhancing our integrated understanding of urban water resources and water partitioning in urban green spaces is crucial for developing an evidence base
needed to underpin sustainable water management.

  The urban water cycle is highly complex and comprises natural and engineered system components that alter the partitioning and routing of precipitation (Gessner et al. , 2014; McGrane , 2016). The spatial distribution of contrasting urban land cover with surfaces of different permeabilities forms a highly heterogeneous mosaic that impacts infiltration and subsurface flow paths (Fletcher et al. , 2013; Pataki et al. , 2011a; Schirmer et al. , 2013). Urban soils are often compacted, disturbed by the preferential
routing of subsurface infrastructure (pipes, cables, etc.) and contain construction-based fill materials and rubble (Endreny , 2005). Additionally, vegetation types of different heights, microclimates and water requirements are often heterogeneously distributed across these complex, anthropogenically-altered landscapes (Nouri et al. , 2013). Consequently, the ecohydrological function of urban green spaces likely differs from more natural vegetation (McGrane , 2016; Pataki et al , 2011b) and process understanding from rural areas may not be transferable (Vico et al. , 2014). For example, urban forests are often a mix of native
and non-native species (Pataki et al , 2011b), and regularly have to adapt to harsh environments with limited water availability where surface sealing is extensive (Bijoor et al. , 2011; Vico et al. , 2014) and a modified radiation and thermal regime from shading by tall buildings (Asawa et al. , 2017). Impervious cover, artificial irrigation and the *UHI* can extend the length of the growing season (Zipper et al. , 2016) and increase urban plant water requirements, thereby posing a risk for dry-season water stress (Zipper et al. , 2017).

Though recent studies have contributed to an improved understanding of urban ecohydrology, there are still considerable knowledge gaps. While volumes of urban storm drainage and water consumed and imported are usually well known, quantifying more natural processes contributing to urban water partitioning remains challenging (Pataki et al. , 2011a). This includes the need for better quantification of *ET* from contrasting vegetation communities comprising urban green infrastructure (Pataki et al. , 2011a; Pataki et al , 2011b; Nouri et al. , 2013) and a better understanding of urbanization impacts on spatio-temporal
variations in infiltration, sub-surface flow paths and groundwater recharge (Schirmer et al. , 2013; McGrane , 2016; Pataki et al. , 2011a). Quantifying these impacts to sustainably manage urban vegetation types and their distribution and placement in the surrounding environment will be crucial for adapting urban water management to meet the water demand of green spaces during projected climate warming and changing precipitation patterns (Nouri et al. , 2013, 2016; Pataki et al , 2011b; Vico et al. , 2014; Zipper et al. , 2017).





A useful tool to disentangle water fluxes and constrain flow paths in complex landscapes is the application of isotopic tracers. Stable isotopes in water behave conservatively and can provide valuable insights into water cycling at various spatio-temporal scales (e.g. Clark and Fritz , 1997; Kendall and McDonnell , 1998). In recent years, they have increasingly been applied to understand water partitioning at the soil-plant-atmosphere interface (Sprenger et al. , 2016, 2019b), particularly the effects of seasonality and different soil and vegetation types (e.g. Geris et al. , 2015; Sprenger et al. , 2017; Oerter and Bowen , 2019). In

drought-sensitive rural areas near Berlin in northern Germany, field-based isotope and hydroclimatic measurements (Kleine et al. , 2020) and integration of tracer data into process-based models (Douinot et al. , 2019; Smith et al. , 2020) revealed higher *ET* losses and lower groundwater recharge under forest than grassland cover. However, comparable studies are scarce in urban areas, where the application of isotope tracers remains a major research frontier (Ehleringer et al. , 2016). One of the first studies using isotopes in urban green spaces suggested groundwater and irrigation as water sources for trees in Los Angeles

(Bijoor et al. , 2011). Others found that trees in the Salt Lake Valley utilized irrigation water during the growing season, though contributions of preceding winter precipitation could always be detected (Gómez-Navarro et al. , 2019). Oerter and Bowen (2017) used *in-situ* isotope measurements in irrigated urban soils in Utah and observed partitioning of soil water into mobile and immobile "pools". While these studies demonstrate the potential of isotope tracers in urban applications, there remains a need to address research gaps in understanding water partitioning not only by urban trees but under the complex mosaic of

vegetation types distributed across urban landscapes.

To address this, we undertook a study in Berlin, Germany; a city characterized by a high cover of urban green and blue spaces, with ~20 % of forest/public green space and 7 % surface waters (SenUVK , 2019a). Berlin's water demand is met by local resources (Limberg , 2007; Möller and Burgschweiger , 2008) and climate projections indicate warmer and drier conditions within the next century, accompanied by an increasing *UHI* (Langendijk et al. , 2019). Therefore, improved knowledge of urban

water partitioning is essential to optimise future benefits from the city's water resources and adapt to more integrated land and water management strategies. Our work focused on plot-scale research at an ecohydrological observatory in Berlin-Steglitz, which provides examples of the main functional urban vegetation types (grassland, shrub and trees) and an opportunity to assess how they affect ecohydrological partitioning. Our specific objectives are to monitor these different urban soil-vegetation units by i) quantitatively assessing ecohydrological partitioning through hydrometric and hydroclimatic measurements, ii)

determining the isotopic composition of precipitation and soil water to trace ecohydrological partitioning effects, iii) inferring water ages and travel times of water in the unsaturated zone and iv) discussing wider implications for upscaling of these findings.

## 2  Study site

Germany's capital Berlin covers 891 km$^2$ with a population of 3.64 million (Amt für Statistik Berlin-Brandenburg, 2020;

Figure 1). Located within the Northern European plain, Berlin's flat topography was formed during the Pleistocene glaciation and primarily consists of Quaternary deposits of unconsolidated sediments (Stackebrandt and Manhenke , 2010). The NE-SW trending sand and gravel deposits of the Berlin-Warsaw-Glacial-Spillway are surrounded by elevated plateaus of Barnim and





Teltow in the North and South, covered by subglacial till (Limberg and Thierbach , 1997; Limberg , 2007). Groundwater flow in Berlin's ~150 m thick freshwater aquifer of Tertiary-Holocene age (Limberg and Thierbach , 1997) is directed from the

plateaus towards the rivers Spree and Havel in the glacial valley (velocities ~10-500 m/yr; Limberg, 2007). Berlin meets its water demand through a semi-closed water cycle, where groundwater is abstracted from local storage and bank filtration, while treated sewage water is discharged back into the surface waters (Limberg and Thierbach , 1997; Limberg , 2007; Möller and Burgschweiger , 2008).

Recharge is low due to high area-weighted $ET$ of 367 mm/yr (60 % of precipitation), plus 152 mm/yr evaporation from

surface waters (SenStadtWoh , 2019). Long-term mean annual rainfall (1981-2010) from weather stations of the German Weather Service (DWD) is 525-602 mm and long-term mean annual air temperature 9.3-10.2 °C (DWD , 2020b). Berlin's land use includes 34 % sealed urban surfaces (SenStadtWoh , 2017) but also extensive urban green spaces, including forests (18.1 %), public parks (12.2 %) and agricultural areas (4.2 %; SenUVK , 2019a). Berlin has a high coverage of surface water bodies (Gerstengarbe et al. , 2003) accounting for ~7 % of the city area (SenUVK , 2019a).

The Steglitz urban ecohydrological observatory (SUEO) is located on the Teltow plateau, in the SW of the city (Figure 1). The surrounding district is covered by residential areas and roads (55 %), forest (24 %), water bodies (11 %) and public green space (9 %; SenUVK , 2019b). The SUEO itself comprises a ~8000 m² large research garden, at an elevation ~45 m a.s.l. (SenStadt , 2010b), with groundwater levels 10-15 m below surface (b.s.; SenStadt , 2010a). Buildings and some older trees were established > 100 years ago, though the current garden dates to the 1950s when the Technical University (TU) Berlin

started using the site (Bornkamm and Köhler , 1987). Recently, further adjustments to the land cover have been made through management of the research garden and installation of a 40 m eddy flux tower as part of the Urban Climate Observatory (UCO) of the TU Chair of Climatology for long-term observations of atmospheric processes in cities (Figure 1). The premises are now covered by buildings (~17 %); green spaces, including grassland (~16 %), shrub (~7 %) and trees (~39 %); and semi-permeable or sealed pathways and parking spaces (~16 %). Soils can be characterized as medium silty/loamy sands (SenStadtWoh , 2018).

They reflect anthropogenic impacts, as the naturally occurring subglacial till is covered by 50-180 cm of debris, sandy materials and a humus layer from long-term intensive cultivation and gardening (Bornkamm and Köhler , 1987).

## 3   Methods and data

Monitoring was carried out from March 2019 to March 2020, with particular focus on the growing season (April to October). Climate data was available from Berlin-Dahlem, ~1500 m west of the site (Precipitation ($P$), air temperature ($T_{air}$), relative

humidity ($RH$), vapor pressure; DWD, 2020a) and the TU UCO eddy flux tower (radiation fluxes, wind speed, $ET$ estimates; Figure 1). Groundwater level data was available from the Berlin Senate (SenUVK , 2020).

Three generic urban vegetation types (grassland, shrub and trees) were selected as representative urban soil-vegetation plots (Figure 1, Table 1). For soil moisture monitoring, CS650 Reflectometers (Campbell Scientific, Inc.; accuracy ±3 % for volumetric water content ($VWC$)) were installed at 10-15 cm ($VWC_{12.5}$), 40-50 cm ($VWC_{45}$) and 90-100 cm ($VWC_{95}$) in each

plot (Figure 1), with duplicate sensors at each depth. Sensors were connected to CR300 Dataloggers in ENC8/10 enclosures





(Campbell Scientific, Inc., Logan, USA). For sap flow, a FLGS-TDP XM1000 Sap Velocity Logger System (Dynamax Inc, Houston, USA) measuring temperature differences between heated sensors (Granier , 1987) was installed at 1.5 m height within six representative urban trees (Figure 1, Table 2). Dependent on tree height and age, two or four sets of sensors were installed in each cardinal direction, with more in older, larger trees. $P$ was sampled daily for isotope analysis using a 3700

Sampler (Teledyne Isco, Lincoln, USA). To prevent evaporation, 1.5 cm of paraffin oil was added to each bottle and occasional samples of < 1.5 mm were rejected in case of exaggerated fractionation effects. For soil water isotope sampling, eight monthly campaigns were conducted from April-November 2019. For each, samples were taken at three locations respectively under grassland, shrubs and trees (Figure 1). Duplicate soil cores were taken from 0-10 cm, 10-20 cm, 40-50 cm and 80-90 cm depth at each location using a soil auger. Samples of ~250 cm$^3$ volume were filled into bags (WEBAbag, Silver Range, Weber

Packaging, Germany), immediately sealed, avoiding air inclusions, and stored in a thermally isolated box. Groundwater and surface water were sampled seasonally for isotope analysis from October 2018 to July 2019 across the whole city (Kuhlemann et al. , 2020). Sampling included typical local, groundwater-fed urban streams (e.g. the Wuhle in the east of Berlin) and a groundwater observation well ~2.5 km NE of the site; these were used in comparison to the SUEO samples for context.

Filtered $P$, groundwater and surface water samples were analysed by Cavity Ring-Down Spectroscopy with a L2130-i Iso-

topic Water Analyzer (Picarro, Inc., USA). Four lab standards were used for linear correction and standards of the International Atomic Energy Agency (IAEA) for calibration. Results were expressed in $\delta$-notation with Vienna Standard Mean Ocean Water (VSMOW). Mean analytical precision was 0.05 ‰ standard deviation ($SD$) for $\delta^{18}$O and 0.16 ‰ $SD$ for $\delta$D.

Soil samples were analysed using the direct equilibrium method (Wassenaar et al. , 2008). First, additional bags were filled with 10 mL of three liquid lab standards (with duplicates). Second, all bags were inflated with dry air, welded and stored for

~48 hours to equilibrate. Third, the vapour phase was analyzed using the Picarro L2130-i by inserting a needle attached to a tube into the bags through the silicon. Standards were measured at the beginning, middle, and end of each run. Criteria for plateau detection during analysis were $SD$ H$_2$O < 100 ppm, $SD$ $\delta^{18}$O < 0.35 and $SD$ $\delta$D < 0.55. Analytical precision was mean $SD$ of 0.14 ‰ and 0.34 ‰ for $\delta^{18}$O and $\delta$D, respectively. Selected bags were re-measured after 2-4 weeks for gas matrix correction (Grahler et al. , 2018). Samples were subsequently oven-dried at 105°C for 24 hours and weighted to determine

their gravimetric water content. Samples with < 3 g of water were excluded from analysis (Hendry et al. , 2015).

For sap flow, individual sensor values for each tree were averaged, converted to sap flux velocity ($u$) in mm/h (Granier , 1987) and summed up to daily totals. Potential evapotranspiration ($PET$) was estimated using the FAO Penman-Monteith method (Allen et al. , 1998). Both $u$ and $PET$ were then normalised (to $u_{norm}$ and $PET_{norm}$, respectively) by subtracting the mean over the study period from the individual daily values and dividing by $SD$.

At each site, $VWC$ of duplicate sensors was averaged hourly. For the growing period, a mass balance approach was applied for a first approximation of $ET$. As the site topography was flat and the study period dry and consistent with the soil moisture data, we assumed no percolation below 50 cm and negligible lateral flow. At each site, daily storage change was calculated as $\Delta S = VWC_{12.5} \cdot h_{12.5} + VWC_{45} \cdot h_{45}$ with $h$ as depth of the soil layer. Daily $ET$ was then estimated as $ET_{calc}(mm) = \Delta S - P$. Occasional small negative daily $ET_{calc}$ values resulting from $P$ inputs that did not infiltrate to sensors were assumed zero. Daily

$ET_{calc}$ was then aggregated to weekly sums. The procedure was repeated for $ET$ from the eddy flux tower at 30 m height. For



comparison of accumulated $ET$, monthly $ET_{calc}$ at the individual sites was summed over the growing period and a weighted mean was calculated considering the fractional distribution of vegetation types at the site. Weekly means were computed for environmental variables ($VWC_{12.5}$; vapour pressure deficit ($VPD_{air}$) calculated from $T_{air}$ and $RH$; and net radiation ($R_n$) calculated from short- and longwave fluxes in 2 m height) to explore linear correlation with weekly $ET_{calc}$ and $u_{norm}$. From

daily $P$ isotopes, a local meteoric water line ($LMWL$) was calculated by amount-weighted least square regression (Hughes and Crawford , 2012). For all isotope samples, *deuterium (d-) excess* was calculated as $d = \delta D - 8 \cdot \delta^{18}O$ (Dansgaard , 1964). To compare soil isotope data with depth, geometric means were calculated from plot and depth replicates. Mean values for soil profiles at the individual sites and sampling campaigns were compared to mean $ET_{calc}$, $T_{air}$, $VPD_{air}$, $R_n$ and $P$ isotopes in the month before or weeks in between each sampling campaign through linear regression. Mean values of seasonally sampled

groundwater and surface water of a local stream (Kuhlemann et al. , 2020) were calculated to compare to $P$ and soil water isotopes.

   To obtain a first approximation of soil water ages at different sites and depths, stable isotopes of $P$ and soil water were used to calculate the fractions of young water ($F_{yw}$) by sine-wave fitting of seasonal cycles (von Freyberg et al. , 2018). Strongly fractionated samples from the upper two soil layers were identified by comparing to the isotope values of incoming $P$ and

excluded from analysis if the soil isotopic values were above the maximum $P$ isotopic value. Mean transit times ($MTT$) were calculated by lumped convolution method (McGuire and McDonnell , 2006). Amount-weighted weekly $P$ isotope means were used as input with a 1-year-spin-up-period. Shape ($\alpha$, range 0.001-5) and scale ($\beta$, range 1-50) parameters were estimated for a gamma transfer function by maximizing the Kling-Gupta-Efficiency ($KGE$; Gupta et al. , 2009) of estimated monthly soil isotopes to the measured monthly soil water isotope data.

## 175   4   Results

### 4.1   Ecohydrological partitioning of water under different generic vegetation communities

Monitoring followed the 2018 summer drought which affected much of Central Europe and below-average $P$ in the winter of 2018/2019. Compared to the long-term mean (1981-2010; DWD , 2020b), 2018 was + 1.6 °C warmer and had a $P$ deficit of 232 mm (or 39%) in Berlin-Dahlem (DWD , 2020a). At the start of the study from March-May, mean daily $T_{air}$ was ~10°C and

$P$ was low ~15 mm/month (Figure 2a,b). From mid-May to mid-June, $T_{air}$ increased to ~18°C and several heavy convective $P$ events totalled >100 mm (Table 3, Figure 2a,b), with the majority of the large events around the third soil sampling with 48 mm (11$^{th}$ June) and 13.6 mm (12$^{th}$ June; DWD , 2020a). Conditions remained warm (~18-22°C) and dry until late September, with most $P$ occurring in high-intensity convective events (Table 3, Figure 2a,b). The remaining period until March 2020 was characterized by lower $T_{air}$ and more frequent, low-intensity $P$. Overall, 2019, similar to 2018, was warmer and drier than

long-term average with a $P$-deficit of 85 mm (14%) and mean annual $T_{air}$ +1.7°C (DWD , 2020a, b).

   Variable $T_{air}$ and $P$ during the study period inevitably impacted soil water storage. Under grassland, $VWC_{12.5}$ was ~27 % in March 2019 decreasing to < 10 % during summer with transient increases after $P$ (Figure 2e). Prolonged re-wetting in October returned $VWC_{12.5}$ to March 2019 levels with minor variability thereafter. At the other sites (Figure 2f-g), $VWC_{12.5}$





was lower generally, with 15 % (shrub) and 25 % (trees) in March, decreasing during dry periods to 2 % in June/July (shrub)

and 4 % in September (trees). Wetness increased again to ~20 % (shrub) and 25 % (trees) over the winter (Figure 2f-g). At all sites, $VWC_{12.5}$ of duplicate sensors was in a similar range. $VWC_{45}$ dynamics were "damped" in comparison. Under grassland, $VWC_{45}$ declined over summer, from 23 % in March 2019 to < 10 % from May onwards. The increase to ~24 % started a few days later than in the upper soil in October (Figure 2e). Under shrub, patterns were similar, with $VWC_{45}$ declining from 11 % in March 2019 to ~1 % throughout the summer and back to 11 % in October (Figure 2f). Under trees, $VWC_{45}$ was 25 % in

March 2019 and slowly declined to 15 % throughout the summer with no marked response to any $P$. It only started to increase to ~20 % in February 2020 (Figure 2g). Temporal variations in $VWC_{95}$ were lowest. Values continuously declined over summer with no response to $P$. Under grassland, $VWC_{95}$ was 19-28 % (Figure 2e). Under shrub, $VWC_{95}$ was higher than at shallower depths (19-24 %; Figure 2f). Under trees, $VWC_{95}$ was low (7-16 %), with a high discrepancy between duplicate sensors (Figure 2g). $VWC_{95}$ only started to increase again in January (grassland, shrub) and March 2020 (trees). Despite some variation, the

groundwater level in the closest well (500 m SE of the SUEO) continuously declined from 9.2 m b.s. in March 2019 to 9.5 m b.s.in March 2020 (Figure 2h), suggesting no net recharge during the study period.

Daily $u_{norm}$ (Figure 2c) showed a marked increase following the start of the growing season in April 2019 as trees came into leave. Values were highest from May-July with daily variability and negative troughs coinciding with larger $P$ events. From August-October, variability remained but $u_{norm}$ decreased, with short-term decreases around $P$ events. After slight variability

in October, rates permanently decreased after leaf fall. $PET_{norm}$ (Figure 2c) showed similar seasonality, with highest rates in June-July. However, during some of the highest $PET_{norm}$ increases, $u_{norm}$ did not respond.

Weekly $ET_{calc}$ ranged from < 5 mm/week under dry, cool conditions to ~40 mm/week after heavy $P$ events (Figure 2d). At the start of the growing period in April, $ET_{calc}$ was ~10 mm/week and highest in the grassland, but after leaf-out, the tree site was higher for some weeks in May and June. Increases in $ET_{calc}$ after large $P$ events were especially pronounced at the tree

site, while grassland $ET_{calc}$ remained highest during dry periods in July and August (Figure 2d). Shrub patterns were variable, with $ET_{calc}$ lowest at the start of the growing season and from July to October, but intermediate from May to July and October and in response to $P$ events (Figure 2d). Accumulated $ET_{calc}$ (Table 4) by October was slightly higher at the tree site (416 mm compared with 414 mm for grass). $ET_{calc}$ at the shrub site was lower with 344 mm. $ET_{calc}$ variations roughly resembled the dynamics measured by the eddy flux tower, although those were more damped (Figure 2d) and totaled only 285 mm over the

same time period. Area-weighted summertime (May-October) $ET_{calc}$ for the vegetation community was 351 mm.

Highest correlations with environmental variables (Figure 3, Table 5) were observed between $u_{norm}$ and $R_n$, and less strongly between $u_{norm}$ and $VWC_{12.5}$, $VPD_{air}$ and $T_{air}$. Correlation between $u_{norm}$ and $VWC_{12.5}$ was negative while all others were positive. Significant positive correlations were also observed between grassland $ET_{calc}$ and $R_n$, $VPD_{air}$ and $T_{air}$. Correlations between $R_n$, $VWC_{12.5}$, $VPD_{air}$ and $T_{air}$ and $ET_{calc}$ at the shrub and tree sites were low.





### 4.2 Ecohydrological partitioning under different urban soil-vegetation units inferred from the isotopic composition of precipitation and soil water

*P* isotopes were generally depleted in winter and more enriched in summer. The range was –17.3 to –0.3 ‰ for $\delta^{18}$O, –131.3 to –12.7 ‰ for $\delta$D and –10.4 to 15.7 ‰ for *d-excess* (Figure 2a). Amount-weighted *LMWL* during our sampling period (Figure 4) was $\delta D = 7.82 \pm 0.26 \cdot \delta^{18}O + 6.31 \pm 1.25 (R^2 = 0.974)$. Soil water samples under grassland showed the widest range and some surface layer samples deviated substantially from the *GMWL* and *LMWL* (Figure 4). However, across the entire soil profile, isotopes were least variable and more enriched under trees. Groundwater and surface water from local urban streams were similar to each other, while most soil water samples were more enriched and plotted further up the *GMWL* and *LMWL*.

Soil water isotopic composition generally became more depleted with increasing depth (Table 6). The upper soil layers under grassland were more enriched compared to shrub and trees, while the lower soil layers were most enriched beneath trees. Mean negative *d-excess*, indicating evaporative losses (Dansgaard , 1964), was observed in the upper soil at the grassland site (Table 6), where $R_n$ and $T_{air}$ significantly influenced the *ET* rates (Figure 3, Table 5), but not at the shrub and tree sites where *d-excess* remained positive.

Soil water isotopes also showed strong temporal variation. In mid-April, when *P* was low but the soils still wet (Figure 2), values were depleted, especially under shrub, and resembled the isotopic signal of incoming *P* at 0-10 cm, while values at 10-20 cm were even more depleted (Figure 5, Figure 6). *D-excess* was more negative at the grassland site, coinciding with higher $ET_{calc}$ (Figure 2d). Following little *P* and increasing $T_{air}$ and $u_{norm}$, *VWC* had decreased by May (Figure 2) and soil water became more enriched, especially at 0-10 cm, while *d-excess* decreased throughout the profile, especially under grassland (Figure 5, Figure 6). While $T_{air}$ and $u_{norm}$ remained high over summer, the large convective *P* event preceding the June sampling led to higher *VWC* and $ET_{calc}$, especially at the tree site (Figure 2), and a more enriched isotopic composition at 0-20 cm. *D-excess*, however, became more positive in the upper soil at all sites, overprinting previous signals of fractionation and displacing waters with lower *d-excess* to depth (Figure 5, Figure 6).

Through the warm, dry July-September period, *VWC* remained low (Figure 2) and the isotopic composition of the upper soil remained enriched but became depleted with depth (Figure 5, Figure 6). *D-excess* in the upper soil was strongly negative at the grassland (Figure 5, Figure 6) where $ET_{calc}$ was slightly higher (Figure 2d). *P* events in August temporarily increased *VWC* (Figure 2e-g) and slightly moderated the fractionation effects (Figure 5, Figure 6). Importantly, it is notable from Figure 6 that from April through July, the isotopic signature of the upper soil (0-10 cm) always moved into the direction of incoming *P* in the weeks preceding the respective samplings. From August through September, however, the isotopic composition of the upper soil became increasingly more enriched than the incoming *P*, especially under grassland (Figure 6). After $T_{air}$, $u_{norm}$ and $ET_{calc}$ decreased and more frequent *P* started in October (Figure 2), soil water isotopes in November were more depleted. However, the deeper soil, especially at 40-50 cm, remained more enriched (Figure 5, Figure 6). Correlations between mean monthly $\delta^{18}$O and $\delta$D of the soil profiles and selected environmental variables in the month before or weeks in between the samplings are shown in Figure 7 and Table 7. Under grassland, soil isotopes showed a strong positive correlation with $T_{air}$,





and lower with $VPD_{air}$. Under trees, soil isotopes showed significant positive correlations with $R_n$, $VPD_{air}$, $T_{air}$ and incoming $P$ isotopes. Under shrub, correlations were weaker, only a positive correlation with $T_{air}$ was evident.

## 4.3 Water ages and travel times of water in the unsaturated zone under different urban vegetation units

Estimated $F_{yw}$ of soil water at the different sites and depths indicate that under grassland, where isotopes and *d-excess* suggested highest evaporative losses over the summer (Figure 5, Figure 6), the upper soil was dominated by young water < 8 weeks old (Figure 8). $MTT$s were < 6 weeks in the upper 20 cm, with good $KGE$ fit (0.39-0.93; Figure 8). A similar pattern was estimated for shrub. Under trees, where $VWC_{12.5}$ was low (Figure 2g) and isotopes indicated less pronounced evaporative enrichment (Figure 5, Figure 6), older water already contributed ~35 % in 10-20 cm depth. $MTT$s of < 8.3 weeks ($KGE$ fit 0.71-0.94) in the upper 20 cm were slightly higher than at the grassland and shrub. In the mid-profile at 40-50 cm depth, $F_{yw}$ remained high (> 70 %) under shrub, where $VWC_{45}$ was lowest (Figure 2f), while only 20-33 % of young water could be observed under grassland and trees (Figure 8). Similarly, $MTT$s were lower under shrub (17-18 weeks, $KGE$ fit 0.8) than under grassland (23-29 weeks, $KGE$ fit 0.44-0.62) and trees (21 to 40 weeks, $KGE$ fit 0.6-0.8). In 80-90 cm, $F_{yw}$ was low at all sites, especially under grassland (5-11 %; Figure 8) where $VWC_{95}$ was higher (Figure 2e), but also at the drier shrub and tree sites (11-18 %). $MTT$s were substantially longer than shallower depths, especially under trees (50-59 weeks) and shrub (49-62 weeks). Due to the lack of variability and the limited observation period, estimates at depth were more uncertain, with $KGE$ fit of 0.13-0.54 at all sites.

## 5 Discussion

### 5.1 Quantitative assessment of ecohydrological partitioning under different urban soil-vegetation units

Through our integrated plot-scale study during and after the exceptional summers of 2018 and 2019, we gained novel insights into ecohydrological partitioning in urban green spaces under dry, warm conditions which will likely become more common in future. The quantitative assessment of urban $ET$ patterns under a mosaic of green spaces, a key challenge in urban ecohydrology (e.g. Nouri et al. , 2013; Pataki et al. , 2011a; Pataki et al , 2011b), showed that "green water" fluxes in the growing period increased in the order shrub < grassland < trees. However, these findings need to be interpreted cautiously as our simple approach to estimating $ET$ through a plot-scale water balance approach may not fully account for spatial heterogeneity in soil moisture distribution and have uncertainties in accounting for deep percolation or capillary rise (Nouri et al. , 2013). Nevertheless, the general similarity of $ET_{calc}$ dynamics to values independently measured by the eddy flux tower indicate that it provided a reasonable first approximation. Higher green space $ET_{calc}$ than flux tower's estimates was expected, as the tower's 30 m height integrates a wider footprint of mixed urban (impermeable) surfaces that can increase surface runoff and decrease evapotranspiration (e.g. Endreny , 2005; Fletcher et al. , 2013; Schirmer et al. , 2013). Throughout the growing season, differences in ecohydrological partitioning under contrasting urban vegetation types were evident, though in some ways counterintuitive. Estimated water use by trees could be reasonably expected to be somewhat higher than grassland (Douinot et





al. , 2019; Smith et al. , 2020), which in turn might be expected to use less water than shrubs. At the grassland site, shading from
surrounding trees was limited and the soil was only covered by the grass sward and patchy moss ground level. Consequently,
with limited interception, incoming $P$ could directly infiltrate and drive the rapid soil moisture dynamics, whilst simultaneously
sustaining transpiration. Similarly, the sparse soil cover enhanced atmospheric exposure for evaporation at the soil surface when
$T_{air}$, $R_n$ and $VPD_{air}$ were high. In contrast, at the tree site, soil was covered by a ground layer of ivy and leaf litter and shaded
by an almost-closed canopy during the growing season. It is likely that much higher interception losses and transpiration by
trees and the understory, the latter reflected by $u_{norm}$, contributed to higher $ET$ rates at this site, leading to drier soils, less
responsiveness to $P$ and longer time lags until re-wetting of the deeper soil in autumn. Inter-sensor variation of $VWC_{95}$ under
trees likely reflects heterogeneity in subsurface texture, as intercalations of sandy and loamy materials were present throughout
the site. Over most of the study period, the shrub site exhibited intermediate hydrological responses to grassland and trees,
e.g. regarding $VWC$ and $ET_{calc}$. However, accumulated $ET$ losses were lowest. This may imply that more water reaches the
soil under shrub than under trees, as interception and transpiration losses from shrubs with a more open canopy and shallower
rooting are lower, but less water directly re-evaporates from the surface than at the grassland site, as some soil cover of ivy
and leaf litter is present. Overall, summertime area-weighted accumulated $ET_{calc}$ of 351 mm for the mixed urban vegetation
community at our site exceeded the sum of incoming $P$ (308 mm; DWD , 2020a) but remained lower than summertime
$PET$ (360 mm) and the annual average area-weighted $ET$ estimates of 367 mm/a (60 % of $P$) for the whole city of Berlin
(SenStadtWoh , 2019). Dependence of seasonal $u_{norm}$ (as a proxy for transpiration) on $R_n$, $VPD_{air}$ and $T_{air}$ is in agreement with
previous observations in urban trees that showed temporal sap flux variability is largely driven by variations in vapor pressure
deficit and photosynthetically active radiation (Asawa et al. , 2017; Pataki et al , 2011b). By selecting a mixed urban tree
assemblage of different tree ages and species, our approach likely integrates the heterogeneity in urban green spaces (cf. Nouri
et al. , 2013). Though some larger $P$ events temporarily increased $VWC$, the simultaneous increase in $RH$ and decrease in $R_n$ and
$VPD_{air}$ caused the transpiration rates to temporarily decrease, explaining negative troughs and correlation. Low dependency of
transpiration rates on soil water content, despite limited $P$, is in contrast to low-energy headwater catchments (Wang et al. ,
2017). This may indicate that transpiration rates at our site showed a certain resilience against prolonged drought periods and
depletion of soil moisture, which would coincide with the rural study east of Berlin following the 2018 drought (Kleine et al. ,
2020). However, decreased $u_{norm}$ during times of highest $PET_{norm}$ would be consistent with the trees conserving water resulting
from moisture stress from the dry sub-soil. This may at least in part also explain the $ET$ losses under trees being less than the
grassland, if low soil moisture limits $ET$. This may also be the result of "memory effects" of the extreme drying in summer
2018 and lower than average re-wetting in the following winter. In addition, the delivery of rainfall in intense convectional
events may limit the time the canopy is wet, with the low radiation and high humidity at such times limiting interception losses
compared to more upland, windy sites (e.g. Soulsby et al. , 2017).



### 5.2 Isotopic composition of precipitation and soil water and its indications for ecohydrological partitioning under different urban vegetation types

The *LMWL* of Berlin-Steglitz was close to those previously reported for Germany and Berlin (Stumpp et al. , 2014). The measured soil water isotopic composition largely supports inference from the hydrometric measurements, but provides more nuanced insights into sources, movement and mixing of stored waters. Over the growing season, changes in soil water isotopes with depth reflected the general pattern of infiltrating *P* becoming more enriched after evaporative losses in the upper 30 cm of soil, while the fractionation signal diminishes with depth as infiltrating *P* mixes with soil waters, damping seasonal variability (Sprenger et al. , 2016). Most pronounced isotopic enrichment and negative *d-excess* under grassland support a pattern of higher soil evaporative losses.

In April/May, more depleted values at 10-20 cm depth at all sites would be consistent with stored water from winter *P* prior to sampling. In contrast, the negative *d-excess* in the upper 10 cm already indicates the effect of evaporative fractionation (Sprenger et al. , 2019a). Subsequent incoming *P* and soil evaporation, especially at the grassland site, strongly influenced the isotopic signal from May through July, though soil water in the upper 10 or even 20 cm was persistently more enriched than incoming *P* in the second half of the growing season. This indicates that by August, despite temporary re-wetting by some larger *P* events, insufficient *P* infiltrated for the soil water to reflect its isotopic signature. This complements recent work in an irrigated urban forest in western USA, which showed that towards the end of the growing season even irrigation was insufficient to replenish soil water storage, and trees "switched" to using deeper, older soil waters (Gómez-Navarro et al. , 2019). Though the isotopic composition of deeper soil layers moved in the direction of groundwater, the high depth of the groundwater table makes any influence through hydraulic redistribution, as recently observed by Oerter and Bowen (2019), unlikely, though a contribution of deeper soil water or groundwater through deep root water uptake from larger trees cannot be ruled out. By the end of November, the more enriched waters had percolated to 40-50 cm depth, but both $VWC_{95}$ and the more enriched isotopic signature in 80-90 cm demonstrate that infiltrating water still hadn't reached this depth, despite more frequent *P* from early October.

The overall more enriched isotopic composition of soil water under trees might point towards a contribution of more enriched throughfall (cf. Geris et al. , 2015; Sprenger et al. , 2017). However, *d-excess* remained high throughout the profile and values were only more enriched compared to the other sites in the lower 40-90 cm, implying no percolation of more fractionated waters. Rather, there seems to be a "mismatch" between soil water in the upper 0-20 cm and soil water in the lower 40-90 cm under trees. Stronger correlation between the isotopic composition of incoming *P* and soil water under trees, along with higher *d-excess*, indicate that canopy and soil cover may preserve the infiltrating *P* signal from direct re-evaporation. Therefore, despite lower net precipitation, the isotopic composition of soil water under trees was most strongly influenced by the isotopic signal of incoming *P* and limited evaporation losses. Assuming that after the dry summer of 2018 percolation to 40-90 cm was similarly late as it was after 2019, when *VWC* only started to increase again towards spring 2020, the more enriched values at depth may be explained by a "memory effect" of recharge from the summer of 2018 under trees, while more infiltrating water over the winter had already replaced or mixed with this water at the grassland and shrub sites. This would be consistent





with recent observations of decoupled hydrological systems by an *in-situ* study in an irrigated urban landscape garden, where

evaporation and irrigation determined highly variable seasonal isotope patterns in the upper 15 cm of soil, while the soil below 20 cm was only hydraulically connected to the upper soil during wetter periods (Oerter and Bowen , 2017). Although a similar study in the rural east of Berlin did not observe a strong memory effect after the 2018 drought as a result of rapid mixing with new rainfall, there was some evidence for a displacement of non-evaporated, more enriched waters from summer to greater depth over the winter of 2018/2019 (Kleine et al. , 2020).

**5.3  Preliminary assessment of water ages and travel times of water in the unsaturated zone under different urban vegetation types**

Higher soil evaporation and shallow root water uptake at the grassland and shrub likely contributed to the predominance of young water and low *MTT* estimates for water stored at 0-20 cm. Greater contributions of older water and slightly higher *MTT* under trees strengthen the hypothesis of longer turnover through interception losses and vegetation water use. The shrub site

now shows a distinct pattern, with a higher fraction of young water with lower *MTT* stored at 40-50 cm. Though this likely reflects a combination of lower interception and less direct evaporation, causing more young water to percolate to this depth, the low $VWC_{45}$ does not fully support this. Proportions of young water predicted at 80-90 cm were low; and indeed hydrometric data suggests that this would be associated with recharge in winter. It is likely that deeper roots under trees and shallower roots under shrub mostly take up older water, thereby increasing the influence of young waters and replenishing *VWC*s in autumn

and winter, a pattern previously observed by Smith et al. (2020). Contributions of older waters, i.e. previous winter recharge to midsummer transpiration, were also observed in trees across Switzerland (Allen et al. , 2019) and in soil and stem waters of irrigated urban forests in the western USA (Gómez-Navarro et al. , 2019).

While hydrometric patterns were in agreement with previous studies in rural catchments of NE Germany, estimated *MTT* and $F_{\text{yw}}$ were not always consistent. Though Douinot et al. (2019) found higher and younger recharge under grass than

under forest, the differences were much greater over a 10-year period. This may link to the greater longevity of that analysis period and resulting lower uncertainty of age estimates. Similarly, Kleine et al. (2020) and Smith et al. (2020) independently reported older water under grassland than under trees. However, this was primarily linked to more silty, water retentive soils under grassland, while soil properties were more consistent between sites in our study.

**5.4  Wider implications**

Many previous studies on urban vegetation have been conducted in semi-arid areas where urban green space is irrigated (e.g. Gómez-Navarro et al. , 2019; Oerter and Bowen , 2017; Pataki et al , 2011b; Nouri et al. , 2019). The absence of irrigation in our current study provided an opportunity to observe more "natural" vegetation water demands and ecohydrological partitioning and can therefore be used to inform strategies for irrigation needs in future. This is especially relevant as our study was carried out following the warmest year in German recorded history (Friedrich and Kasper , 2019), a relatively dry winter and

consecutive dry summer with heavy convective *P* events, thereby providing a first assessment on how urban green spaces may



react to future climate changes. As long-term trends (1981-2010) showed that in the Berlin area > 50 % of $P$ falls in spring and summer alone (DWD , 2020c), the enhanced frequency of dry summers will inevitably lead to increasing water shortages.

Though urban forests can provide enhanced cooling benefits (e.g. Gunawardena et al. , 2017), recent studies showed increasing emission of latent heat in grassland rather than forest during drought conditions in Europe (Lansu et al. , 2020). While such
large-scale findings may not be easily transferable to plot-scale, urban site, isotope tracers in our study revealed higher soil evaporation under urban grassland, though tree transpiration and interception lead to similarly high $ET$ rates over the growing season. However, pronounced depletion of soil moisture, longer recovery times and slower turnover of soil water under urban trees raise the question, how the water supply for urban trees can be maintained if prolonged drought periods increasingly occur in the future. This is especially important as the $UHI$ can increase $PET$ and vegetation demand in urban areas compared to
rural surroundings (Zipper et al. , 2017). Upscaling these findings means that, in coming years, irrigation management is likely to be increasingly needed to support urban trees where soils are freely draining, followed by urban grasslands, while shrubs may be more resilient. Taking such aspects into consideration for selecting suitable plant and tree species in the future will be crucial for sustainable management of urban green spaces and limiting a city's water footprint (Nouri et al. , 2019; Vico et al. , 2014). As particularly the right combination of urban "green" and "blue" space can provide effective cooling mechanisms and
ecosystem benefits (Gunawardena et al. , 2017; Hathway and Sharples , 2012), Berlin with its high vegetation and water cover has exceptional potential for better use of these features. Despite these preliminary insights, it is clear that water partitioning in urban green spaces is complex, and more work is needed over longer timescales for a deeper understanding of ecohydrological partitioning under contrasting urban vegetation and upscaling these findings to the city scale. For more quantitative understanding of seasonal water cycling under the different vegetation types at our site, future work will integrate our field-based
data into a process-based model (cf. Douinot et al. , 2019). This will also help resolve the green water fluxes into estimates for interception, transpiration and soil evaporation (Smith et al. , 2020). Additionally, upscaling will require more extensive data collection across Berlin. Eventually, this will lead to a more complete picture of how heterogeneously distributed urban vegetation alters urban water partitioning, using approaches that can be transferred to many other urban areas.

## 6 Conclusions

Through our plot-scale study of seasonal water cycling in Berlin-Steglitz, we gained insights into ecohydrological partitioning under different types of urban green spaces during prolonged dry periods and heavy precipitation events. Our results indicate that contrasting urban vegetation cover can significantly affect infiltration patterns and $ET$ rates, as seen in variations in soil moisture regimes, isotopic signals and transit times. Despite high soil evaporation losses, urban grassland allowed more direct percolation of rainwater and maintained higher moisture levels. Interception losses and vegetation water use contributed to
similarly high $ET$ under urban trees. Resulting from the high water demand of urban trees, soils at the tree site were driest, and suggested a decoupled hydrological system with slower turnover times and recharge from the previous summer still present at higher depth. Shrubs seemed to exhibit lower soil evaporative losses compared to the grassland site and a higher moisture content through lower interception losses and root water uptake compared to the tree site; making this vegetation type potentially





more resilient to persistent drought conditions. These insights can contribute to a better adaption of species-specific irrigation
strategies in the future. However, more research is needed to upscale these findings to the city-scale and gain more profound
insights into the prevailing processes by integrating our field data into process-based ecohydrological models.

*Data availability.* The data that support the findings of this study are available from the corresponding author upon reasonable request.

*Author contributions.* The study was designed by LK, DT, and CS. Field work and data collection was undertaken by LK. Data was analysed
by LK, with ongoing discussion and inputs from DT, CS and AS. LK prepared the draft manuscript, to which subsequently all authors
contributed and edited.

*Competing interests.* The authors declare that they have no conflict of interest.

*Acknowledgements.* We thank the German Research Foundation (DFG) for funding this project as part of the Research Training Group
"Urban Water Interfaces (UWI)" (GRK 2032) and the Einstein Foundation for the support as part of the project "Modelling surface and
groundwater with isotopes in urban catchments (MOSAIC)". We are especially thankful to our colleagues of the TU Berlin Ecology Depart-
ment for providing access to their property and assistance for site selection, in particular B. Seitz, and to the Department of Climatology,
especially D. Scherer and F. Meier, for providing the UCO climate data. Further, we thank our colleagues E. Brakkee, L. Lachmann, N. Weiß,
C. Marx, L. Kleine, W. Lehmann, H. Dämpfling, D. Dubbert and M. Gillefalk for assistance in the soil sampling and sensor installation; A.
Wieland, N. Weiß, H. Dämpfling, J. Freymüller and S. Jordan for their help with the precipitation sampling and D. Dubbert for help with
the isotope analysis. Finally, we thank the Berlin Senate Department for the Environment, Transport and Climate Protection for providing
groundwater data and well access.



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



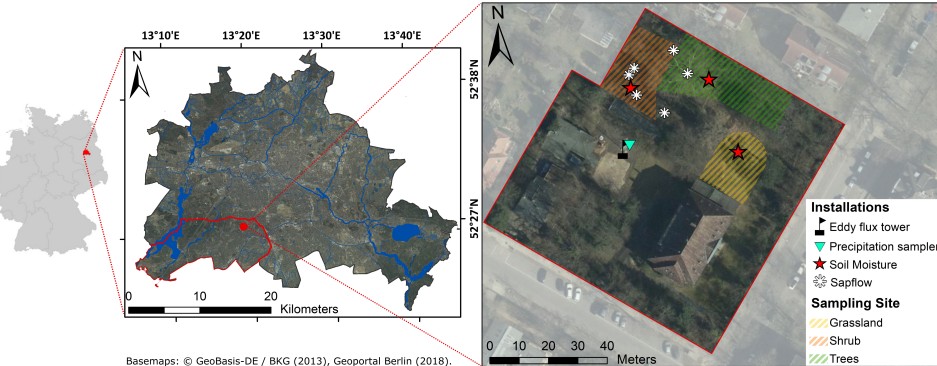

**Figure 1.** Location of Berlin within Germany (left); location of the district of Steglitz-Zehlendorf and the SUEO (red) with Berlin's surface waters in blue (middle); and the SUEO (right) with vegetation plots and installations of soil moisture and sap flow measurements, precipitation sampler and eddy flux tower.





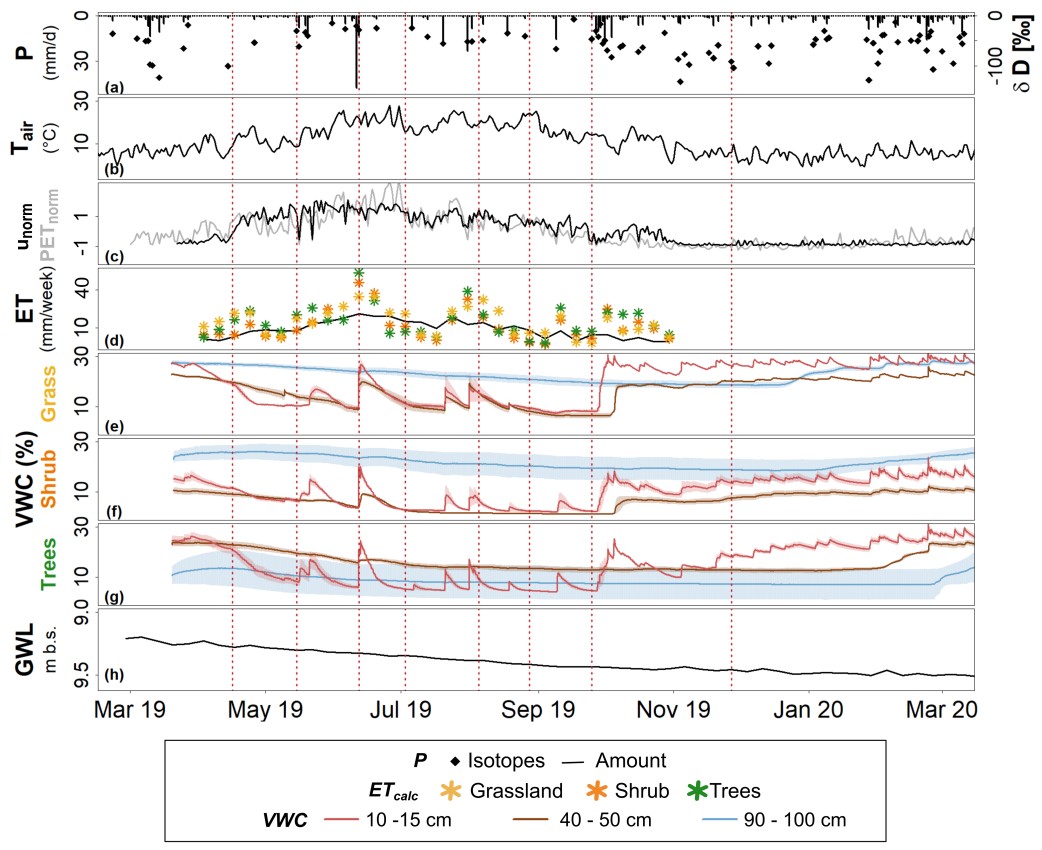

**Figure 2.** a) daily $P$ sums (DWD, 2020a) and isotopic composition, b) mean daily $T_{air}$ (DWD, 2020a), c) daily $u_{norm}$ and $PET_{norm}$ (grey) d) weekly $ET_{calc}$ and $ET$ of the eddy flux tower, e-g) soil $VWC$ at different depths and sites, and h) mean weekly groundwater level (GWL) near the SUEO (SenUVK, 2020). Red dashed lines mark the days where soil samples were taken for monthly soil water isotope analysis.





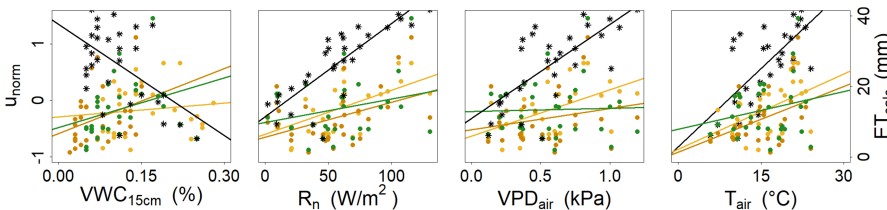

**Figure 3.** Linear correlation between weekly $u_{norm}$ (asterisk) and weekly $ET_{calc}$ at the grassland (yellow), shrub (orange) and tree (green) sites and climatic variables.





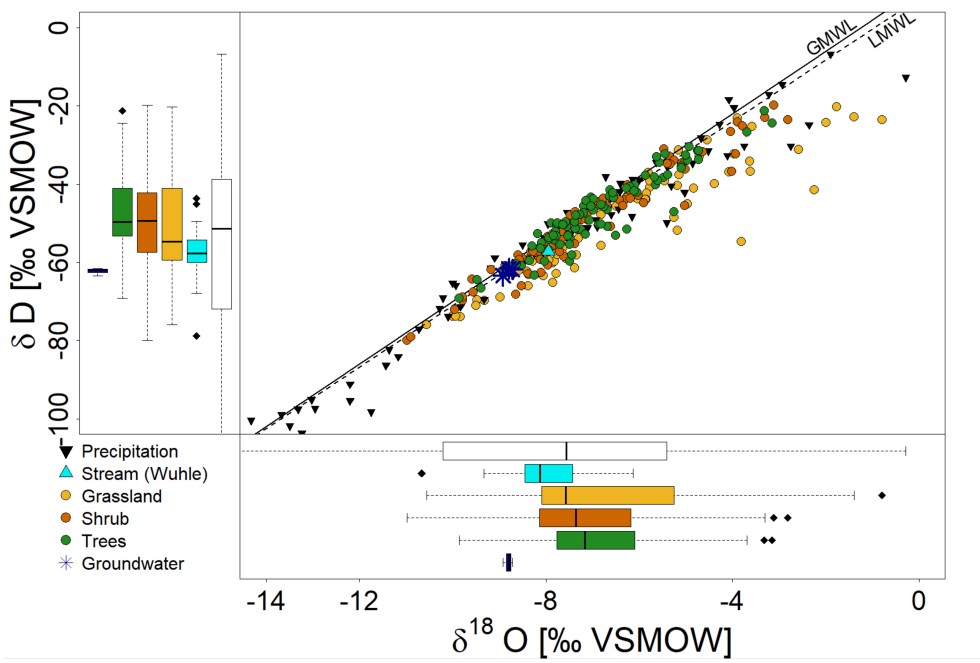

**Figure 4.** Dual isotope plot showing the isotopic composition of *P* and soil water isotopes at the grassland, shrub and tree sites, along with the mean isotopic composition of surface water of a local, groundwater-fed, urbanized stream (Wuhle) and groundwater sampled (~2.5km north of the SUEO (Kuhlemann et al., 2020).





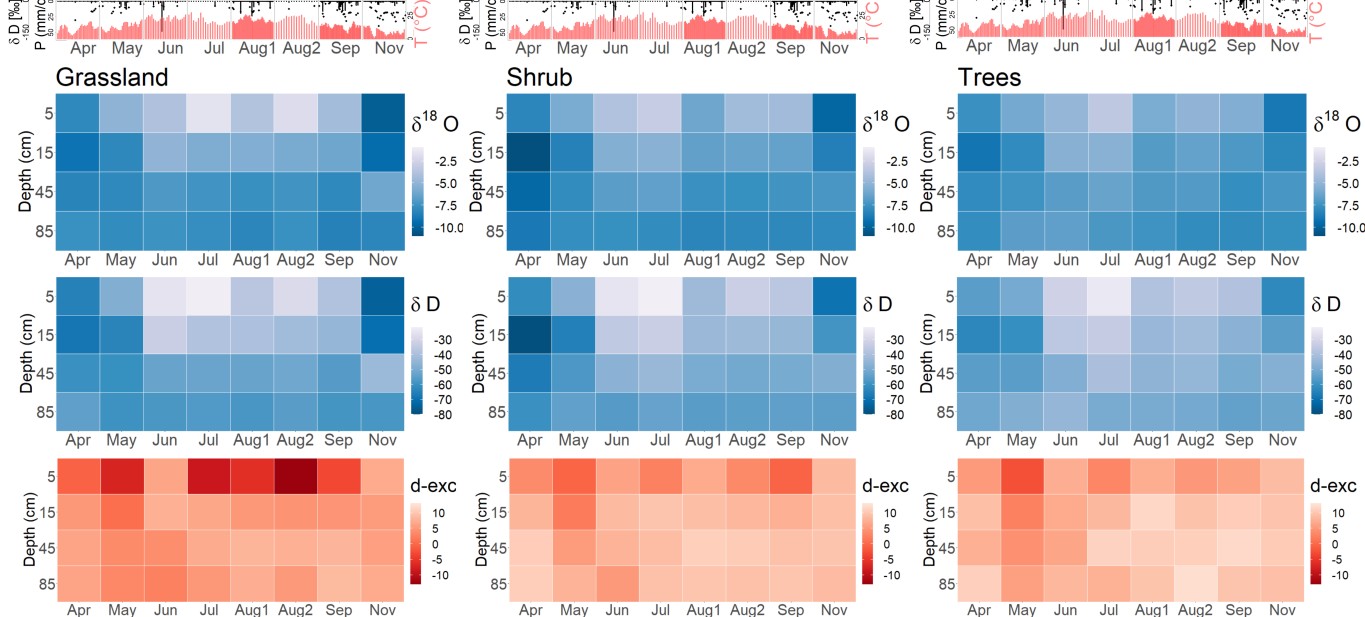

**Figure 5.** Heatmaps showing the isotopic composition of the different soil layers during the monthly sampling campaigns (abbreviations see Table 3) in ‰ VSMOW. Climate parameters (top; DWD, 2020a) mark the daily $T_{air}$ and $P$ during the sampling period.





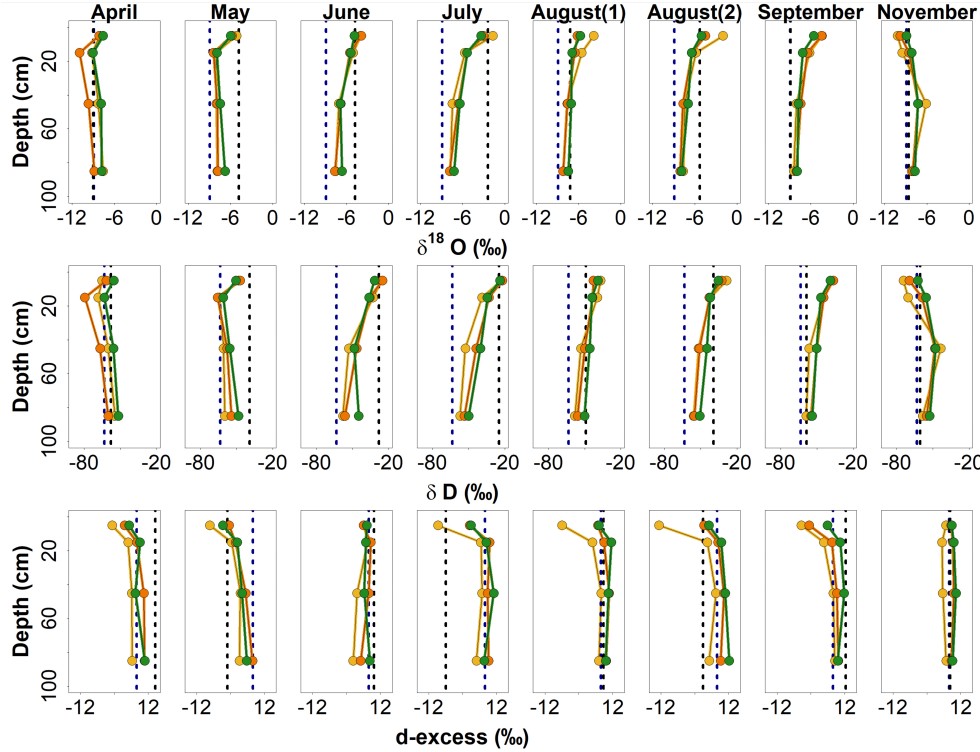

**Figure 6.** Isotopic depth profiles showing geometric means at different depths at the grassland (yellow), shrub (orange) and tree site (green) during the monthly sampling campaigns. Dashed lines marking the mean isotopic composition of groundwater (blue) and weighted *P* mean in the month before or weeks in between the sampling campaigns are given for reference.





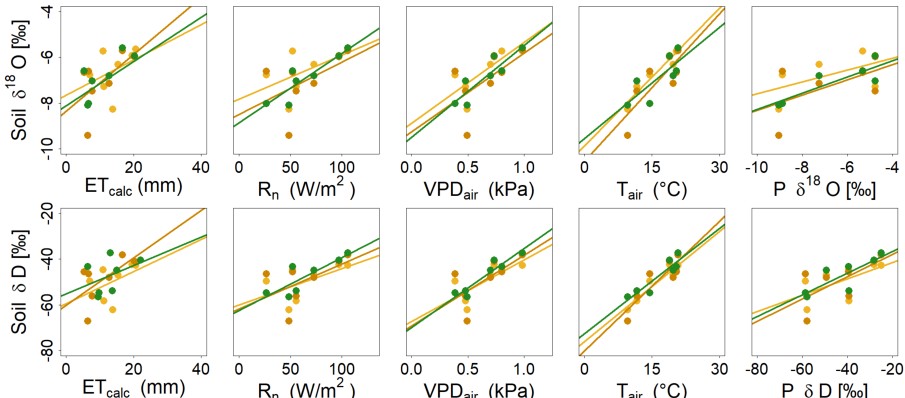

**Figure 7.** Linear correlations at the grassland (yellow), shrub (orange) and tree site (green) between mean anlysed soil water isotopic composition at the individual sampling dates and mean values of environmental parameters in the month before or weeks in between the respective samplings.





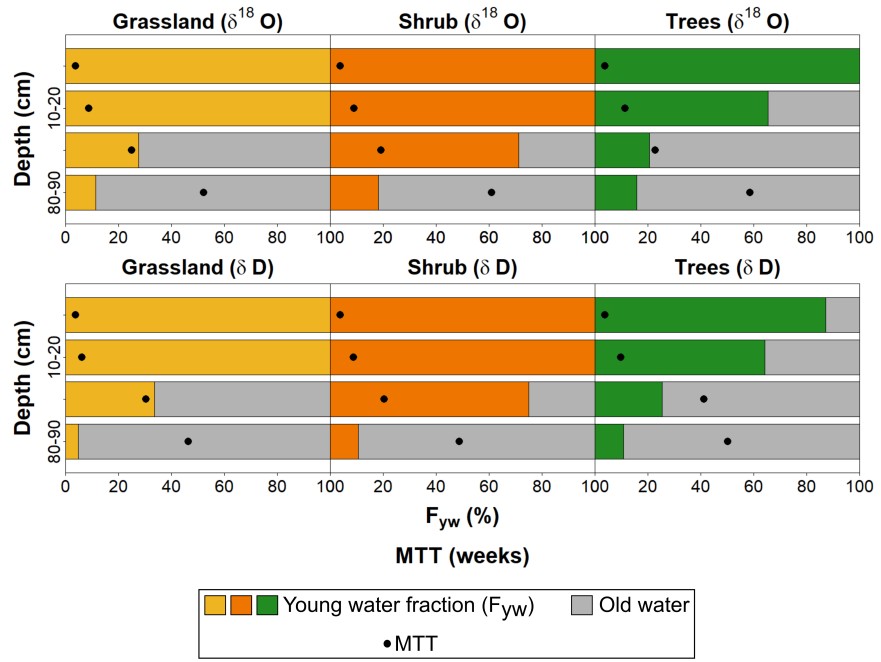

**Figure 8.** $F_{yw}$ and $MTT$s at different sites and depths during the growing period of 2019.





**Table 1.** Main plant species covering the three sampling plots and their approximate heights and estimated mean LAI ranges over the study period at the different vegetation plots.

| Site | Vegetation cover | Vegetation height | Estimated LAI |
|---|---|---|---|
| Grassland | *Arrhenatherum elatius* <br> Moss | < 0.5 m | 2-3 |
| Shrub | Leaf litter | < 0.5 m | 2-4 |
| | *Clematis, Hedera helix* | | |
| | *Rubus armeniacus* | 0.5 to 1 m | |
| | Young trees (*Acer platanoides, Acer pseudoplatanus*) | 2 to 3 m | |
| Trees | Leaf litter | < 0.5 m | 3-5 |
| | *Hedera helix, Allium ursinum* | | |
| | Various tree species of more than 100 years of age (including oak, birch, plane, maple, elm, pine, chestnut, ash, lime tree, larch) | 10 to > 20 m | |
| | Some shrub and smaller trees (*Prunus padus, Sambucus nigra, Euonymus europaeus, Symphoricarpos, Ligustrum vulgare, Anemone nemorosa, Mahonia*) | 0.5 to 1.5 m | |




**Table 2.** Characteristics of trees and sensors of the sap flow installation.

| Tree species | Diameter at breast height (cm) | Number of sensors | Sensor type |
|---|---|---|---|
| | 8.9 | 2 | TDP10 |
| Maple *(Acer platanoides, Acer pseudoplatanus)* | 10.5 | 2 | TDP30 |
| | 14.0 | 2 | TDP30 |
| Elm *(Ulmus glabra)* | 18.5 | 2 | TDP30 |
| Plane *(Platanus x hybrida)* | 111.4 | 4 | TDP30 |
| Oak *(Quercus robur)* | 67.8 | 4 | TDP50 |





**Table 3.** Climate parameters (DWD , 2020a) and *VWC* under the different vegetation units in the month before or weeks in between the individual soil sampling campaigns.

| Sampling | | 1 (Apr) | 2 (May) | 3 (Jun) | 4 (Jul) | 5 (Aug1) | 6 (Aug2) | 7 (Sep) | 8 (Nov) |
|---|---|---|---|---|---|---|---|---|---|
| **Time period** | | 16.3.-16.4. | 16.4.-15.5 | 15.5.-12.6. | 12.6.-3.7. | 3.7.-5.8. | 5.8.-28.8. | 28.8.-25.9. | 25.9.-27.11. |
| $T_{air}$(°C) | Mean | 8.12 | 11.70 | 17.96 | 21.50 | 19.35 | 20.09 | 17.65 | 9.25 |
| | SD | 3.12 | 3.13 | 4.13 | 3.31 | 3.28 | 2.46 | 4.00 | 4.22 |
| *P* (mm) | Sum | 11.90 | 13.20 | 103.10 | 2.80 | 63.70 | 19.40 | 41.80 | 121.20 |
| *VWC* **Grassland** | Mean | 24.30 | 18.34 | 16.50 | 19.16 | 15.35 | 14.46 | 12.98 | 20.28 |
| | SD | 1.27 | 1.38 | 1.56 | 2.51 | 1.62 | 1.27 | 1.60 | 2.37 |
| *VWC* **Shrub** | Mean | 16.41 | 13.89 | 12.90 | 12.33 | 9.24 | 8.51 | 8.27 | 12.20 |
| | SD | 0.55 | 0.74 | 1.58 | 2.21 | 0.61 | 0.37 | 0.45 | 1.19 |
| *VWC* **Trees** | Mean | 20.14 | 15.30 | 12.57 | 11.74 | 9.78 | 9.12 | 8.98 | 11.71 |
| | SD | 0.50 | 1.80 | 1.64 | 1.95 | 0.91 | 0.40 | 0.55 | 1.18 |



**Table 4.** Accumulated $ET_{calc}$ over the growing period of 2019.

| | $ET_{\text{calc}}$ (mm) | | |
|:---:|:---:|:---:|:---:|
| | **Grassland** | **Shrub** | **Trees** |
| **April** | 62.00 | 33.82 | 51.76 |
| **May** | 112.81 | 80.18 | 116.25 |
| **June** | 209.16 | 171.21 | 203.51 |
| **July** | 262.58 | 216.60 | 244.16 |
| **August** | 335.26 | 261.36 | 305.78 |
| **September** | 364.32 | 288.11 | 345.08 |
| **October** | 413.60 | 343.92 | 415.59 |





**Table 5.** $R^2$ and p-values of the parameters used for the correlation plots of $u_{norm}$ and $ET_{calc}$ (Figure 3).

| | | $VWC_{15}$ | | $T_{air}$ | | $VPD_{air}$ | | $R_n$ | |
|---|---|---|---|---|---|---|---|---|---|
| | | $R^2$ | p-value | $R^2$ | p-value | $R^2$ | p-value | $R^2$ | p-value |
| $u_{norm}$ | | 0.27 | 3.00E-03 | 0.44 | 4.56e-05 | 0.49 | 1.33e-05 | 0.71 | 2.49E-09 |
| | Grassland | 0.01 | 0.54 | 0.25 | 4.06E-03 | 0.29 | 1.70E-03 | 0.39 | 1.70E-04 |
| $ET_{calc}$ | Shrub | 0.10 | 0.08 | 0.16 | 0.03 | 0.05 | 0.23 | 0.20 | 0.01 |
| | Trees | 0.10 | 0.08 | 0.05 | 0.24 | 1.00E-03 | 0.86 | 0.09 | 0.10 |





**Table 6.** Amount of samples (n) with measured isotopic composition of *P* and soil water under the three soil-vegetation units for different sampling depths.

**Precipitation**

|  | $\delta^{18}O$ (‰) | $\delta D$ (‰) | d-exc. (‰) |
|---|---|---|---|
| n | 78 | 78 | 78 |
| Mean | -8.23 | -58.74 | 7.06 |
| SD | 3.58 | 27.65 | 5.21 |

**Soil water**

|  | Grassland | | | Shrub | | | Trees | | |
|---|---|---|---|---|---|---|---|---|---|
|  | $\delta^{18}O$ (‰) | $\delta D$ (‰) | d-exc. (‰) | $\delta^{18}O$ (‰) | $\delta D$ (‰) | d-exc. (‰) | $\delta^{18}O$ (‰) | $\delta D$ (‰) | d-exc. (‰) |
| **0-10 cm** | | | | | | | | | |
| n |  | 23 |  |  | 22 |  |  | 23 |  |
| Mean | -5.01 | -42.83 | -2.69 | -5.70 | -41.61 | 4.59 | -6.00 | -43.07 | 5.54 |
| SD | 2.83 | 18.57 | 7.57 | 2.21 | 16.22 | 5.03 | 1.69 | 12.50 | 3.69 |
| **10-20 cm** | | | | | | | | | |
| n |  | 23 |  |  | 22 |  |  | 23 |  |
| Mean | -6.92 | -50.53 | 6.14 | -7.16 | -49.50 | 9.29 | -7.26 | -49.74 | 9.79 |
| SD | 1.74 | 14.75 | 3.39 | 1.72 | 14.76 | 3.22 | 1.21 | 10.18 | 3.33 |
| **40-50 cm** | | | | | | | | | |
| n |  | 22 |  |  | 17 |  |  | 22 |  |
| Mean | -7.34 | -53.90 | 6.47 | -7.50 | -51.22 | 11.02 | -7.18 | -48.52 | 10.37 |
| SD | 1.33 | 6.69 | 8.25 | 1.04 | 8.36 | 2.91 | 0.72 | 7.00 | 3.68 |
| **80-90 cm** | | | | | | | | | |
| n |  | 18 |  |  | 14 |  |  | 18 |  |
| Mean | -7.72 | -57.65 | 5.69 | -8.03 | -55.96 | 10.97 | -7.12 | -48.06 | 9.63 |
| SD | 1.03 | 2.28 | 8.60 | 0.60 | 3.94 | 3.00 | 1.13 | 7.93 | 2.52 |






**Table 7.** Correlations between analysed soil water isotopic composition and mean environmental variables 4 weeks prior to the sampling.

|  |  | $VWC_{15}$ | | $T_{air}$ | | $VPD_{air}$ | | $R_n$ | | $P$ isotopes | |
|---|---|---|---|---|---|---|---|---|---|---|---|
|  |  | $R^2$ | p-value | $R^2$ | p-value | $R^2$ | p-value | $R^2$ | p-value | $R^2$ | p-value |
| $\delta^{18}O$ | **Grassland** | 0.15 | 0.38 | 0.93 | 5e-4 | 0.60 | 0.04 | 0.32 | 0.18 | 0.44 | 0.10 |
|  | **Shrub** | 0.35 | 0.16 | 0.65 | 0.03 | 0.35 | 0.16 | 0.28 | 0.22 | 0.45 | 0.10 |
|  | **Trees** | 0.26 | 0.24 | 0.62 | 0.04 | 0.79 | 7e-3 | 0.81 | 6e-3 | 0.85 | 4e-3 |
| $\delta D$ | **Grassland** | 0.19 | 0.32 | 0.91 | 8e-4 | 0.54 | 0.06 | 0.33 | 0.17 | 0.38 | 0.14 |
|  | **Shrub** | 0.38 | 0.14 | 0.78 | 8e-3 | 0.44 | 0.10 | 0.31 | 0.19 | 0.44 | 0.11 |
|  | **Trees** | 0.17 | 0.37 | 0.85 | 3e-3 | 0.89 | 1e-3 | 0.72 | 0.02 | 0.68 | 0.02 |