# Peer review of "Using soil water isotopes to infer the influence of contrasting urban green space on ecohydrological partitioning."

_Hydrology and Earth System Sciences, 2020_

## Referee Comment (RC1) · Anonymous Referee #1 · 30 Oct 2020

This manuscript presents evapotranspiration, soil moisture, isotopic variations, and travel time estimates from isotopic variations in the soil beneath grassland, shrubs, and trees in plots in Berlin during a drought. These are all valid analyses. However, my main concern with this paper is that for some reason the authors motivate this work by discussing green infrastructure, mitigation of flood risk, urban growth, urban water demand, urban soil compaction, and the urban water cycle. All of these topics are quite peripherally related to the actual study that was conducted. The study that was conducted happens to be studying vegetated plots that are located in an urban area, but otherwise there is not evidence that the plots are affected by any of the urban processes that are discussed in the introduction. The plots are not irrigated, which makes

the discussion of urban water demand seem off-topic. (This should be stated clearly up front – only in the discussion is it explicitly stated that the plots do not receive irrigation. The reader should know this immediately). The plots are not used to manage stormwater, which makes the green infrastructure and flood risk mitigation discussion seem off topic. The plots are not obviously affected by urban soil compaction. There are no interactions with urban infrastructure mentioned. It is not clear why the urban ecohydrology study framing is being used since it is not clear what urban process is actually being studied here. There is an incredible amount of data being presented but the motivation is comparatively lacking. Secondly, the discussion section has some text that is unsupported by the work presented. There is no evidence to suggest that irrigation is needed to support the trees that are currently not irrigated (L390-392). Making this sort of unsupported statement can have large water use implications, so this type of statement on increasing or beginning irrigation should be made carefully. Do the researchers believe that the trees will die if not irrigated? There was no data on the tree water stress to support this belief in this manuscript. The goal of urban irrigation is not to maximize ET or plant growth or plant yield but to keep species alive that humans want alive (e.g., urban lawns are irrigated to a level much below well-watered conditions for maximal ET by most residential irrigators because they irrigate based on visual plant stress (DeOreo 2016 – Residential End Use Study)). If the trees are not dying under currently non-irrigated conditions, then what reason is there to suggest that they should be irrigated to maintain them? This is also discussed in L414. Major Comments: L100: State clearly and up front that the vegetation is not irrigated in SUEO. L1: The first sentence of the abstract raises questions and is confusing. Does 'green infrastructure' here refer to urban vegetation? Green infrastructure in other contexts is used to mean green stormwater infrastructure which in many cities is not heavily irrigated and therefore is not a major challenge to balancing domestic and industrial water demand. Also, are many urban areas having challenges in balancing domestic and industrial water demands, or rather in having a limited resource of water for all urban water use (both domestic (does this refer to residential indoor + outdoor or just

residential indoor – I am not familiar with this terminology) and industrial). L300 and throughout: Why is unorm used as a proxy for transpiration? L330 and L349: The authors are comparing their findings to other work looking at urban landscapes with isotopes. However, it doesn't makes sense to me to make these comparisons because the other studies they compare to are irrigated – this seems fundamentally different since the irrigation introduces a different isotopic source and signature into the system and is completing altering the inputs to the soil-vegetation system. It would make more sense to me to compare the results from this study to other studies (non-urban or urban) that look at non-irrigated grassland vs. shrub vs. tree comparisons. L376-378: Why would observations of more natural vegetation water demands be used to inform strategies for irrigation needs in the future? If the vegetation is natural in this study, why have the entire motivation be pointing to urbanization? L377-378: It is inappropriately general to state this so broadly – that these are characteristics of 'urban trees'. This finding is not generalizable to this degree – these findings are currently specific to both the specific location and to the species studied (tree species have quite different ET from each other and relationships to water availability). Minor Comments: L202-203: This is not evident from the plot. A marking at '0' or the absolute value of unorm would help. L212: 'Slightly higher' seems to overstate the difference of only 2 mm – I would interpret this to mean they are effectively the same. What kind of error bars would these values have? Figure 2: The x-axis should have marks for every month. Table 3: Clarify what sampling time period refers to. L224: The widest range for O, not for H. L225: Clarify what across the entire soil profile refers to. L311: What type of memory effects? Storage? Stomata? Vegetation?

---

## Author Comment (AC1) · 2 Nov 2020

We thank reviewer 1 for their comments.

We are pleased that the reviewer recognises that our study presents an "incredible amount of data" and that we carry out a "valid analyses" of these. We also appreciate the limited number of major and minor comments on technical aspects of the paper that we feel that we can easily address in review.

However, the reviewer expresses more fundamental concern about how we contextualise (and then discuss) the motivation behind this work in terms of urban ecohydrology

and urban water management. These concerns revolve around the following issues: (1) the introduction provides a broad background to urban water management issues, many of which, in the reviewer's opinion, are of minor relevance to the study presented (e.g. green infrastructure, urban drainage etc.), making the motivation for the work is unclear. (2) Our study plots happen to be in an urban area and are not subject to many common urban pressures (e.g. receiving storm drainage, compaction etc.). (3) We make inappropriate and unsupported inferences about drought stress on the tree plot and possible need for irrigation. (4) We inappropriately compare the isotopic element or our work to results from irrigation studies in other urban settings. We address these in turn.

1. Context

In retrospect we can see that our introduction is too broad and needs to be shortened and re-focused in revision. However, we were trying to set the issue of water partitioning in urban green space in the broader context of issues for urban water management. In this regard, we, like Berlin's water managers, view "green infrastructure" more broadly than technical structures, to include all green space affecting water partitioning. In this regard, we see our work contributing to the growing (but still limited) number of studies assessing water partitioning in contrasting urban green space. Moreover, a fundamental motivation was to use isotopes as tools to help in this, again because there are so few urban studies. While we see that introduction could cause confusion on the motivation of the work, we would argue that the title of the paper and the objectives the study are actually quite clear.

2. Urban setting

Whilst the reviewer is correct that our plots are not irrigated, compacted or in receipt of storm drains, they are still typical of tree, shrub and grassland in managed urban green space in a major European city. Thus, they are not natural vegetation, have small dimensionality, and are subject to urban climate effects, artificial soil debris etc.

[Figure]

Moreover, they are typical of the majority of Berlin's urban vegetation, in not being irrigated and not in receipt of storm runoff. They are clearly informative urban study sites, that form part of the green infrastructure. We will emphasise these issues on revision.

3. Drought stress on trees

The reviewer is right that we have limited information on this and no proof, though we would highlight that our comments on the water-limitation on trees were suitably circumspect. But we do feel that the issue is worth raising in that it provides a potential explanation why the ET under trees was not significantly greater than form the grassland. The point we are making is that with climate change, vegetation that has been sustainable in the past, may no longer continue to be. However, we will be even more circumspect in revision.

4. Comparison to irrigation studies

The reviewer questions the comparison of our work to other papers where isotopes have been used to assess the effects of urban irrigation. The issue here is that these are some of the very limited number of isotope studies in urban settings that we can actually compare our work to, and so help evaluate the potential of isotopes in urban ecohydrology. Of course, there are a plethora of isotope studies in more natural settings in a wide range of geographical regions, most of which are irrelevant to the study site. Hence, we refer to some of the studies more relevant to Berlin. On revision, we will search the literature for any very recent, potentially relevant studies.

---

## Referee Comment (RC2) · Anonymous Referee #2 · 13 Nov 2020

I largely enjoyed reading this paper by Kuhlemann et al. In fact, I thought it got better as I progressed through the manuscript. The paper describes hydrometric measurements, stable isotope variation and isotope-based estimates of young water fractions and transit times in three plots designed to represent different green spaces in the urban environment. I think that interest in aspects of urban hydrology is growing and that studies like this are needed to continually improve our understanding of processes in heavily human-impacted systems.

There are a few issues with the paper, but in total, I do not think these issues are dire enough to prevent the paper's eventual publication. My suggestion is along the lines of

a "major revision".

From an experimental design standpoint, it is unfortunate that replication is somewhat minimal. There is some within-plot pseudo-replication of certain measurements, but I remain a little hung-up on trying to assess whether the grass plot, shrub plot and tree plot are indeed representative of what one might come across in an urban space and whether the measurements and findings really do represent the breadth of variability that exists naturally. That said, there is a rich dataset here and I do believe that the authors can mostly move forward with what is written. I would suggest that the authority of the writing should however be tempered to match the lack of replication (and therefore the lack of understanding of heterogeneity or representativeness) and at some point, be more explicit in the paper about how replication with the necessary investment into these types of plot studies is not always feasible – and finally, how the lack of replication leads to some unknown uncertainty.

I found the general premise of the writing of the introduction and parts of the discussion/implications to be not as directly related to the work as I would hope. There is significant context given to climate change and irrigation, but I would suggest that the experiment does not hit squarely on either all that well. Irrigation in most temperate urban spaces is not much of an issue. Even if it is in the future, I'm not sure this work is directly transferrable to answering much about that. For climate change, the work does fit well with a drought scenario, but there's no real "change" that is within the design of the study. Perhaps the more direct way forward is to couch the paper more about soil water dynamics in some typical urban greenspace areas under conditions affected by recent drought. This is mostly what is already here and thus, wouldn't be too much of a pivot. The use of stable isotopes, especially through time, is quite novel, and the title of the paper does fit well with a somewhat differently focused introduction. It would not be too much a stretch to wait until later in the discussion to make the fuller climate change and future urban water management. Setting the scene this way up front just does not quite represent the work effectively in my opinion.

The results section is described in a pretty dense way, especially in its first half. It would benefit in readability if the authors tried to synthesize more toward the principal observations and let the figures partly speak for themselves, at least a little bit.

Finally, this is quite squarely an ecohydrological study, but the ecological part of that is somewhat lacking in the discussion. Are all grass-covered, shrub-covered and tree-covered soils in urban systems really expected to act according to the study's observations? What feedbacks might we expect in a changing climate for different vegetation covers? Do we know much about how species composition/community composition might impact on the observations from the study?

Some more specific comments: • Line 2: Is maintaining the water supply for green infrastructure really a particularly serious issue in temperate climate cities? In many circumstances, the purpose of green infrastructure is largely to help control too much water on the surface of the landscape. • Line 6: "effects" of? Vegetation type? • Line 25: is climate "breakdown" really a term in common use? • Line 25: abstractions, or extractions? • Line 117: The study description could use a bit more (couple of sentences) in terms of fundamental experimental design explanation. • Line 141: What silicon? This sentence could use some editing. Are you trying to explain that there was some sort of silicon septum on the bag? • Line 226: How relevant are stream water and groundwater in the context of this study exactly? The "experiment" is more a plot-scale experiment, so this seems to be a brief, but unfocused part of the study. • Line 271: This first discussion paragraph is a pretty long paragraph without a good central theme, but rather quite a few disparate points being made. Could it be broken up a bit to focus the main points better? • Section 5.2: I found this section quite interesting and well explained. It hits me that it would be nice to have an idea of field capacity in order to situate this a bit more closely with deeper percolation and eventually groundwater recharge. This might not be possible, but could it be inferred/estimated from the soil moisture time series maybe? • Line 378-381: For context to some of my earlier comments, I found this sentence to best situate the study

into a discussion of climate change.  c Line 381-382: Something is oddly explained here given summer and spring are indeed 50% of the year.  c Descriptions related to the word "depth" such as at line 412, but I believe maybe elsewhere: it is better not to use higher/lower in relation to depth. Deeper/shallower is easier to understand.  c Figures and Tables: I think these are nicely done. I have a few comments.  c Figure 3: it took me a minute to realize that u-norm is only relevant to the treed plot. I would suggest that the caption needs some editing to more clearly describe what is being regressed against what. Also, given that most of the p-values in the associated Table 5 are not statistically significant, is it actually meaningful to include linear lines of best fit for the insignificant relationships?  c Figure 4: Though clear in the paper, the labelling of this figure could be more specific about the measurements being soil pore water in grassland, shrub, trees - or at least make clear in the caption. Otherwise, one risks quick readers thinking this is isotope information in water within grass, shrub or trees.  c Figure 5 and 6: I think the heatmap is an interesting way of doing this, but I honestly would prefer to see the evolution of the soil profile through time, which I think is exactly what is shown in figure 6. I would like the authors to consider if figure 5 and 6 are too closely representing the same information and whether this should be collapsed into just one figure. Could the heatmap part of figure 5 not just be replaced with the entirety of figure 6 (keeping the top part of figure 5 still)?

---

## Author Response (AR1)

*Dear Prof. Coenders-Gerrits,*

*Thank you for these positive comments and the opportunity to improve our manuscript. Below, we provide responses to the reviewer comments. We have clarified the motivation of the study.*

*Yours sincerely,*
*Lena-Marie Kuhlemann, on behalf of all coauthors.*

**Editor Decision:** Publish subject to revisions (further review by editor and referees) (04 Dec 2020) by Miriam Coenders-Gerrits

*Comments to the Author:*
Both reviewers are rather positive on your study; however they both argue the motivation of your study and especially the need for urban irrigation. I agree with your proposal and looking forward to see it implemented.
*We have now implemented the proposed changes in the revised manuscript. Please find the point-by-point responses below.*

In addition, I would like to advise to move all topics that do not follow 1:1 from the study, but are worth mentioning, to move them to a discussion section.
*We have significantly condensed the introduction; and omitted or moved any topics not directly related to the results of our study or needed for context.*

Furthermore, I request you to follow the data policy of HESS. The current data statement ("The data that support the findings of this study are available from the corresponding author upon reasonable request.") is not compliant. Data should be made publicly available unless there is a clear detailed explanation why this is not possible. Please see for further information: https://www.hydrology-and-earth-system-sciences.net/policies/data_policy.html
*Some of the data are part of current modeling work and can not be made publically available just yet. However, we provide a reference now to the open access data base FRED at IGB,*

*where most of the data sets will be made available. In addition, the author team has extensive expertise in collaborations when colleagues approach us for exchanging data and process-based knowledge. We have revised the data availability statement accordingly.*

**Anonymous Referee #1**

This manuscript presents evapotranspiration, soil moisture, isotopic variations, and travel time estimates from isotopic variations in the soil beneath grassland, shrubs, and trees in plots in Berlin during a drought. These are all valid analyses.

*We thank reviewer 1 for their comments. We are pleased that the reviewer recognises that our study presents an "incredible amount of data" and that we carry out a "valid analyses" of these. We also appreciate the limited number of major and minor comments on technical aspects of the paper and have addressed those accordingly in the review.*

However, my main concern with this paper is that for some reason the authors motivate this work by discussing green infrastructure, mitigation of flood risk, urban growth, urban water demand, urban soil compaction, and the urban water cycle. All of these topics are quite peripherally related to the actual study that was conducted. The study that was conducted happens to be studying vegetated plots that are located in an urban area, but otherwise there is not evidence that the plots are affected by any of the urban processes that are discussed in the introduction. The plots are not irrigated, which makes the discussion of urban water demand seem off-topic. (This should be stated clearly up front – only in the discussion is it explicitly stated that the plots do not receive irrigation. The reader should know this immediately). The plots are not used to manage stormwater, which makes the green infrastructure and flood risk mitigation discussion seem off topic. The plots are not obviously affected by urban soil compaction. There are no interactions with urban infrastructure mentioned. It is not clear why the urban ecohydrology study framing is being used since it is not clear what urban process is actually being studied here. There is an incredible amount of data being presented but the motivation is comparatively lacking.

*In retrospect we can see that our introduction is too broad and needs to be shortened and re-focused in revision. However, we were trying to set the issue of water partitioning in urban*

*green space in the broader context of issues for urban water management. In this regard, we, like Berlin's water managers, view "green infrastructure" more broadly than technical structures, to include all green space affecting water partitioning.*

*We see our work contributing to the growing (but still limited) number of studies assessing water partitioning in contrasting urban green space. Moreover, a fundamental motivation was to use isotopes as tools to help in this, again because there are so few urban studies. While we see that introduction could cause confusion on the motivation of the work, we would argue that the title of the paper and the objectives of the study are actually quite clear.*

*Whilst the reviewer is correct that our plots are not irrigated, compacted or in receipt of storm drains, they are still typical of tree, shrub and grassland in managed urban green space in a major European city. Thus, they are not natural vegetation, have small dimensionality, and are subject to urban climate effects, artificial soil debris etc. Moreover, they are typical of the majority of Berlin's urban vegetation, in not being irrigated and not in receipt of storm runoff. They are clearly informative urban study sites that form part of the green infrastructure. We have emphasized this in the revised paper.*

Secondly, the discussion section has some text that is unsupported by the work presented. There is no evidence to suggest that irrigation is needed to support the trees that are currently not irrigated (L390-392). Making this sort of unsupported statement can have large water use implications, so this type of statement on increasing or beginning irrigation should be made carefully. Do the researchers believe that the trees will die if not irrigated? There was no data on the tree water stress to support this belief in this manuscript. The goal of urban irrigation is not to maximize ET or plant growth or plant yield but to keep species alive that humans want alive (e.g., urban lawns are irrigated to a level much below well-watered conditions for maximal ET by most residential irrigators because they irrigate based on visual plant stress (DeOreo 2016 – Residential End Use Study)). If the trees are not dying under currently non-irrigated conditions, then what reason is there to suggest that they should be irrigated to maintain them? This is also discussed in L414.

*The reviewer is right that we have limited information on this and no proof, though we would highlight that our comments on the water-limitation on trees were suitably circumspect. But we do feel that the issue is worth raising in that it provides a potential explanation why the*

*ET under trees was not significantly greater than form the grassland. The point we are making is that with climate change, vegetation that has been sustainable in the past, may no longer continue to be. We now stress this more in the revised manuscript.*

*Major Comments:*

L100: State clearly and up front that the vegetation is not irrigated in SUEO.
*We have added this information in the revised manuscript.*

L1: The first sentence of the abstract raises questions and is confusing. Does 'green infrastructure' here refer to urban vegetation? Green infrastructure in other contexts is used to mean green stormwater infrastructure which in many cities is not heavily irrigated and therefore is not a major challenge to balancing domestic and industrial water demand. Also, are many urban areas having challenges in balancing domestic and industrial water demands, or rather in having a limited resource of water for all urban water use (both domestic (does this refer to residential indoor + outdoor or just residential indoor – I am not familiar with this terminology) and industrial).
*We have rephrased this in the revised manuscript. Though we already emphasized in our previous comments that we view urban green infrastructure in a broader sense, we now distinguish more clearly between urban technical green structures (e.g. green roofs) vs. more natural urban green spaces (e.g. parks) in the introduction.*

L300 and throughout: Why is unorm used as a proxy for transpiration?
*The calculation of transpiration would have required knowledge about the tree's sap wood area, usually obtained through drill cores and scaled up to a forest stand in terms of transpiration flux (in mm). However, as the property was a small, diverse botanic garden with conservation priorities, we were unable to take these cores. Moreover, given the heterogeneity in stand age and species composition, upscaling would have been highly uncertain. Rather than giving individual (absolute) values, normalizing sap flux velocity provided a more general overview of the transpiration dynamics throughout the growing season not only in one tree or tree species, but in an assemblage of trees typically found in*

*urban areas. We added the purpose of this normalization for both $u_{norm}$ and PET in the method section in the revised manuscript.*

L330 and L349: The authors are comparing their findings to other work looking at urban landscapes with isotopes. However, it doesn't makes sense to me to make these comparisons because the other studies they compare to are irrigated – this seems fundamentally different since the irrigation introduces a different isotopic source and signature into the system and is completing altering the inputs to the soil-vegetation system. It would make more sense to me to compare the results from this study to other studies (non-urban or urban) that look at non-irrigated grassland vs. shrub vs. tree comparisons.

*The reviewer questions the comparison of our work to other papers where isotopes have been used to assess the effects of urban irrigation. The issue here is that these are some of the very limited number of isotope studies in urban settings that we can actually compare our work to, and so help evaluate the potential of isotopes in urban ecohydrology. Of course, there are a plethora of isotope studies in more natural settings in a wide range of geographical regions, most of which are irrelevant to the study site. Hence, we refer to some of the studies more relevant to Berlin. Though we did not find additional studies where isotopes are used in non-irrigated urban green spaces, we included a recent study on the overall drought response of urban vegetation and emphasize more that we additionally compare to studies in rural areas near Berlin, where the vegetation plots are not irrigated.*

L376-378: Why would observations of more natural vegetation water demands be used to inform strategies for irrigation needs in the future? If the vegetation is natural in this study, why have the entire motivation be pointing to urbanization?

*For the context aspect of this study, please refer to our response on this topic in the previous paragraphs. That said, even if more natural vegetation in urban green spaces is not yet heavily irrigated in more temperate regions like Berlin, the increasing frequency of warmer and drier periods will likely require such measures in the future. By observing water use and partitioning in green spaces now, we believe that an improved understanding of these processes can inform on sustainable urban water management and irrigation strategies in the future. We now stress this more in the revised manuscript.*

L377-378: It is inappropriately general to state this so broadly – that these are characteristics of 'urban trees'. This finding is not generalizable to this degree – these findings are currently specific to both the specific location and to the species studied (tree species have quite different ET from each other and relationships to water availability).

*Please see our response to the previous comment.*

*Minor Comments:*

L202-203: This is not evident from the plot. A marking at '0' or the absolute value of unorm would help.

*We added this "0" mark in Figure 2c.*

L212: 'Slightly higher' seems to overstate the difference of only 2 mm – I would interpret this to mean they are effectively the same. What kind of error bars would these values have?

*We rephrased this; the reviewer is right, these differences are not statistically significant given likely uncertainty. For the uncertainty, this is the focus on a forthcoming paper which disaggregates the water fluxes using a physically-based ecohydrological model. This is now mentioned in the revision.*

Figure 2: The x-axis should have marks for every month.

*Thank you for the suggestion. We changed this.*

Table 3: Clarify what sampling time period refers to.

*This was not meant as a single expression, but rather referring to the top two rows: the sampling ($1^{st}$ sampling in April, $2^{nd}$ sampling in May, etc.) and below the time period preceding the respective sampling, which was used for the calculation of the given means/SDs. To clarify this, we added a line between these rows, to separate the information more effectively.*

L224: The widest range for O, not for H.

*Thank you for flagging this up. We added this information.*

L225: Clarify what across the entire soil profile refers to.

*We specified this.*

L311: What type of memory effects? Storage? Stomata? Vegetation?

*We clarified this; we are referring to soil moisture storage.*

**Anonymous Referee #2**

I largely enjoyed reading this paper by Kuhlemann et al. In fact, I thought it got better as I progressed through the manuscript. The paper describes hydrometric measurements, stable isotope variation and isotope-based estimates of young water fractions and transit times in three plots designed to represent different green spaces in the urban environment. I think that interest in aspects of urban hydrology is growing and that studies like this are needed to continually improve our understanding of processes in heavily human-impacted systems. There are a few issues with the paper, but in total, I do not think these issues are dire enough to prevent the paper's eventual publication. My suggestion is along the lines of a "major revision".

*Thank you for these positive comments.*

From an experimental design standpoint, it is unfortunate that replication is somewhat minimal. There is some within-plot pseudo-replication of certain measurements, but I remain a little hung-up on trying to assess whether the grass plot, shrub plot and tree plot are indeed representative of what one might come across in an urban space and whether the measurements and findings really do represent the breadth of variability that exists naturally. That said, there is a rich dataset here and I do believe that the authors can mostly move forward with what is written. I would suggest that the authority of the writing should however be tempered to match the lack of replication (and therefore the lack of understanding of heterogeneity or representativeness) and at some point, be more explicit in the paper about how replication with the necessary investment into these types of plot studies is not always feasible – and finally, how the lack of replication leads to some unknown uncertainty.

*This is a valid point. We tried to cover heterogeneities in soil and vegetation cover by distributing our soil sampling across different locations in the respective vegetation plots and taking duplicate soil moisture measurements and spatially distributed isotope measurements. However, as in most studies, resources limited the degree of replication that was possible. While we agree that more work needs to be done to put our results into a wider context of other parks across Berlin, such work was beyond the scope of this plot-scale study. However, we plan to address this in future research and are currently undertaking similar investigations in different parks across Berlin. This sampling will complement our current study, inform on the representativeness and facilitate an upscaling of our findings to the city-scale. We stress this more in the discussion of the revised manuscript.*

I found the general premise of the writing of the introduction and parts of the discussion/ implications to be not as directly related to the work as I would hope. There is significant context given to climate change and irrigation, but I would suggest that the experiment does not hit squarely on either all that well. Irrigation in most temperate urban spaces is not much of an issue. Even if it is in the future, I'm not sure this work is directly transferrable to answering much about that. For climate change, the work does fit well with a drought scenario, but there's no real "change" that is within the design of the study. Perhaps the more direct way forward is to couch the paper more about soil water dynamics in some typical urban greenspace areas under conditions affected by recent drought. This is mostly what is already here and thus, wouldn't be too much of a pivot.

*Again, this is a valid point and was raised by Reviewer 1. In the revised manuscript, have significantly cut down the implications for general urban water demands in the introduction. Rather, we now stress the distinction between urban areas where water shortages and irrigation of green spaces area already an issue; and, in contrast, more temperate regions where, so far, the irrigation need of urban green has been limited. However, in cities like Berlin, the last years were so exceptionally warm and dry that city planners had to start irrigating urban trees. We therefore believe that irrigation of urban green in more temperate regions will gain importance in the future and that understanding water cycling of non-irrigated urban green spaces in these areas is important, as could help inform on which vegetation types will have higher irrigation needs in the future and therefore facilitate the implementation of sustainable urban irrigation strategies. This is also stressed in the revised*

*manuscript. We now also emphasize the need for longer study periods for a more complete picture.*

The use of stable isotopes, especially through time, is quite novel, and the title of the paper does fit well with a somewhat differently focused introduction. It would not be too much a stretch to wait until later in the discussion to make the fuller climate change and future urban water management. Setting the scene this way up front just does not quite represent the work effectively in my opinion.

*We adapted this aspect in the introduction of the revised manuscript. While we agree that the wider climate change context is not needed here, we, however, stress why our study is important in the context of increasingly warm and dry conditions in temperate regions, as this is directly connected to our study area.*

The results section is described in a pretty dense way, especially in its first half. It would benefit in readability if the authors tried to synthesize more toward the principal observations and let the figures partly speak for themselves, at least a little bit.

*Thank you for this suggestion. We condensed the first half of the results section of the revised manuscript.*

Finally, this is quite squarely an ecohydrological study, but the ecological part of that is somewhat lacking in the discussion. Are all grass-covered, shrub-covered and tree covered soils in urban systems really expected to act according to the study's observations? What feedbacks might we expect in a changing climate for different vegetation covers? Do we know much about how species composition/community composition might impact on the observations from the study?

*These are all interesting questions. However, our current plot-scale study does not yet cover a spatial or temporal extent that would allow a sufficient answer to them. Rather, addressing these questions will require more extensive field work, longer observation periods and potentially also new modelling approaches. We sketch out our plans to take on these approaches in the future in the discussion of the revised manuscript.*

*Some more specific comments:*

Line 2: Is maintaining the water supply for green infrastructure really a particularly serious issue in temperate climate cities? In many circumstances, the purpose of green infrastructure is largely to help control too much water on the surface of the landscape.

*We rephrased this in the revised manuscript.*

Line 6: "effects" of? Vegetation type?

*We rephrased this in the revised manuscript.*

Line 25: is climate "breakdown" really a term in common use? Line 25: abstractions, or extractions?

*This paragraph was removed in the revised manuscript, in order to shorten and focus the introduction.*

Line 117: The study description could use a bit more (couple of sentences) in terms of fundamental experimental design explanation.

*We added more details on the rationale behind the experimental setup throughout the method (and study site) section of the revised manuscript.*

Line 141: What silicon? This sentence could use some editing. Are you trying to explain that there was some sort of silicon septum on the bag?

*Yes. Thank you for pointing this out, this information was apparently lost in our editing process. We have clarified this in the revised manuscript.*

Line 226: How relevant are stream water and groundwater in the context of this study exactly? The "experiment" is more a plot-scale experiment, so this seems to be a brief, but unfocused part of the study.

*We included this information here for context. However, we agree that the information on the isotopic composition of surface water may be of minor importance for this paper, as the plot-scale study site does not include and is not directly adjacent to a larger urban stream as sampled in other areas of Berlin. However, we believe that the isotopic composition of groundwater is relevant for this study, as it allows a comparison between the sampled soil*

*water (especially at 90 cm) and the groundwater at depth, potentially providing information on recharge sources. Therefore, we kept the groundwater values in our plot.*

Line 271: This first discussion paragraph is a pretty long paragraph without a good central theme, but rather quite a few disparate points being made. Could it be broken up a bit to focus the main points better?

*We agree and re-structured and condensed this section in the revised manuscript.*

Section 5.2: I found this section quite interesting and well explained. It hits me that it would be nice to have an idea of field capacity in order to situate this a bit more closely with deeper percolation and eventually groundwater recharge. This might not be possible, but could it be inferred/estimated from the soil moisture time series maybe?

*This is an interesting suggestion. We looked into this but couldn't estimate the field capacity and due to the high heterogeneity and anthropogenic impacts generally found in urban soils we felt not comfortable just transferring values from another site.*

Line 378-381: For context to some of my earlier comments, I found this sentence to best situate the study into a discussion of climate change.

*Noted.*

Line 381-382: Something is oddly explained here given summer and spring are indeed 50% of the year.

*This paragraph was omitted in the revised manuscript.*

Descriptions related to the word "depth" such as at line 412, but I believe maybe elsewhere: it is better not to use higher/lower in relation to depth. Deeper/shallower is easier to understand.

*This is a good point. We changed this terminology throughout the manuscript.*

Figures and Tables: I think these are nicely done.

*Thank you.*

I have a few comments.

Figure 3: it took me a minute to realize that u-norm is only relevant to the treed plot. I would suggest that the caption needs some editing to more clearly describe what is being regressed against what. Also, given that most of the p-values in the associated Table 5 are not statistically significant, is it actually meaningful to include linear lines of best fit for the insignificant relationships?

*Thank you for pointing this out. We removed all linear lines representing statistically insignificant correlations. We further rephrased the figure caption, pointing out that measured $u_{norm}$ only refers to the tree site. As this comment may also apply to the second correlation plot, we also removed lines for insignificant relationships in Figure 7.*

Figure 4: Though clear in the paper, the labelling of this figure could be more specific about the measurements being soil pore water in grassland, shrub, trees - or at least make clear in the caption. Otherwise, one risks quick readers thinking this is isotope information in water within grass, shrub or trees.

*We rephrased the figure caption to emphasize that soil water isotopes refer to the sampled bulk water at different depths under the different vegetation types.*

Figure 5 and 6: I think the heatmap is an interesting way of doing this, but I honestly would prefer to see the evolution of the soil profile through time, which I think is exactly what is shown in figure 6. I would like the authors to consider if figure 5 and 6 are too closely representing the same information and whether this should be collapsed into just one figure. Could the heatmap part of figure 5 not just be replaced with the entirety of figure 6 (keeping the top part of figure 5 still)?

*We understand that these plots present similar information. However, we included both of these plots, as we believe they each stress specific aspects of these results. On the one hand, the temporal development during the growing season and especially the differences in d-excess, indicating evaporative fractionation in the upper soil, are more apparent in the heatmaps in Figure 5. On the other hand, the development in the individual profiles with depth is probably easier to distinguish in the depth profiles of Figure 6. Additionally, this plot provides the perspective of comparing the soil water isotopic composition to incoming*

*precipitation and groundwater. Therefore, we keep both plots in the revised manuscript as they add different insights.*